# Preference Optimization by Estimating the Ratio of the Data Distribution

**Yeongmin Kim**[1]  **Heesun Bae**[1]  **Byeonghu Na**[1]  **Il-Chul Moon**[1,2]
[1]Korea Advanced Institute of Science and Technology (KAIST), [2]summary.ai
{alsdudrla10, cat2507, byeonghu.na, icmoon}@kaist.ac.kr

## Abstract

Direct preference optimization (DPO) is widely used as a simple and stable method for aligning large language models (LLMs) with human preferences. This paper investigates a generalized DPO loss that enables a policy model to match the target policy from a likelihood ratio estimation perspective. The ratio of the target policy provides a unique identification of the policy distribution without relying on reward models or partition functions. This allows the generalized loss to retain both simplicity and theoretical guarantees, which prior work such as $f$-PO fails to achieve simultaneously. We propose *Bregman preference optimization* (BPO), a generalized framework for ratio matching that provides a family of objective functions achieving target policy optimality. BPO subsumes DPO as a special case and offers tractable forms for all instances, allowing implementation with a few lines of code. We further develop scaled Basu's power divergence (SBA), a gradient scaling method that can be used for BPO instances. The BPO framework complements other DPO variants and is applicable to target policies defined by these variants. In experiments, unlike other probabilistic loss extensions such as $f$-DPO or $f$-PO, which exhibit a trade-off between generation fidelity and diversity, instances of BPO improve both win rate and entropy compared with DPO. When applied to Llama-3-8B-Instruct, BPO achieves state-of-the-art performance among Llama-3-8B backbones, with a 55.9% length-controlled win rate on AlpacaEval2. Project page: https://github.com/aailab-kaist/BPO.

## 1 Introduction

Aligning large language models (LLMs) with human feedback has emerged as a promising fine-tuning paradigm to better reflect human preferences [1, 18, 56]. Reinforcement learning from human feedback (RLHF) [13, 47] is a widely adopted alignment method that bridges implicit human preferences and model behaviors. This method typically involves two stages after supervised fine-tuning: (1) training a reward model that captures implicit human preferences, and (2) optimizing LLMs through a reinforcement learning pipeline guided by the learned reward model. This multi-stage RLHF pipeline is computationally intensive and prone to instability, motivating the development of direct preference optimization (DPO) [49] as an alternative approach.

DPO is an alignment method that does not require an auxiliary reward model, thereby improving training efficiency and stability. DPO training reduces to logistic regression on offline preference datasets, following the Bradley–Terry model [9]. Subsequent studies have proposed extensions of the DPO loss functions [55, 62, 67]. From the perspective of distribution matching, $f$-PO [27, 67] reformulates the alignment objective as matching the policy model to the optimal policy defined by DPO, extending the loss using $f$-divergence. While this reformulation retains the target optimality, $f$-PO introduces additional complexity since it relies on a learned reward model and requires Monte Carlo estimation of partition functions. Importantly, its optimality cannot be guaranteed without incurring additional computational cost, as summarized in the third and fourth rows of Table 1.

39th Conference on Neural Information Processing Systems (NeurIPS 2025).

Table 1: Summary of probabilistic DPO extensions. **O**: optimality preservation, **S**: simplicity without extra training cost, **G**: generality for multiple objectives. Refer to Appendix B for baseline details.

| Name | O | S | G | Loss $[\mathbf{S}(\checkmark) : (\mathbf{x}, \mathbf{y}_w, \mathbf{y}_l) \sim p_{\text{data}}]$ |
|---|---|---|---|---|
| DPO [49] | ✓ | ✓ | ✗ | $-\log\sigma\left(\beta\log\dfrac{\pi_\theta(\mathbf{y}_w|\mathbf{x})}{\pi_{\text{ref}}(\mathbf{y}_w|\mathbf{x})} - \beta\log\dfrac{\pi_\theta(\mathbf{y}_l|\mathbf{x})}{\pi_{\text{ref}}(\mathbf{y}_l|\mathbf{x})}\right)$ |
| $f$-DPO [62] | ✗ | ✓ | ✓ | $-\log\sigma\left(\beta f'\left(\dfrac{\pi_\theta(\mathbf{y}_w|\mathbf{x})}{\pi_{\text{ref}}(\mathbf{y}_w|\mathbf{x})}\right) - \beta f'\left(\dfrac{\pi_\theta(\mathbf{y}_l|\mathbf{x})}{\pi_{\text{ref}}(\mathbf{y}_l|\mathbf{x})}\right)\right)$ |
| $f$-PO [67] | ✓ | ✗ | ✓ | $D_f\left(\pi_\theta(\mathbf{y}|\mathbf{x})\,\middle\|\,\dfrac{1}{Z(\mathbf{x})}\pi_{\text{ref}}(\mathbf{y}|\mathbf{x})\exp\left(\dfrac{1}{\beta}r_{\phi^*}(\mathbf{x},\mathbf{y})\right)\right)$ |
| $f$-PO [67] | ✗ | ✓ | ✓ | $f\left(\sigma\left(\beta\log\dfrac{\pi_\theta(\mathbf{y}_w|\mathbf{x})}{\pi_{\text{ref}}(\mathbf{y}_w|\mathbf{x})} - \beta\log\dfrac{\pi_\theta(\mathbf{y}_l|\mathbf{x})}{\pi_{\text{ref}}(\mathbf{y}_l|\mathbf{x})}\right)\right)$ |
| BPO (Ours) | ✓ | ✓ | ✓ | $h'(R_\theta)R_\theta - h(R_\theta) - h'\left(R_\theta^{-1}\right), R_\theta: \left[\dfrac{\pi_\theta(\mathbf{y}_l|\mathbf{x})\pi_{\text{ref}}(\mathbf{y}_w|\mathbf{x})}{\pi_\theta(\mathbf{y}_w|\mathbf{x})\pi_{\text{ref}}(\mathbf{y}_l|\mathbf{x})}\right]^\beta$ |

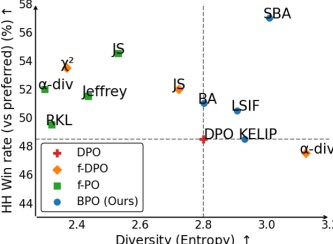

Figure 1: The trade-off between fidelity and diversity across instances from different probabilistic preference optimization frameworks on dialogue generation with Pythia 2.8B.

This paper investigates a generalized DPO loss that enables policy models to match the target policy without incurring additional computational burden. We show that the optimal policy can be expressed without relying on a learned reward model or partition functions by adopting a likelihood ratio perspective. The likelihood ratio of the target policy is a valid estimation target, as the ratio uniquely identifies the target policy distribution. We reformulate DPO as a ratio matching problem between the data preference ratio and the model ratio, and extend the loss using Bregman divergence [10, 53], which we refer to as *Bregman preference optimization* (BPO). BPO generalizes DPO by including it as a special case. We also propose a scaled Basu's power (SBA) divergence, a gradient scaling method for BPO instances. In addition, we show that the proposed loss generalization can be applied in an orthogonal manner to existing DPO variants that can be expressed as logistic regression.

We empirically evaluate several instances of BPO against baselines, measuring both generation fidelity (win rate with GPT-4 judge) and generation diversity. In contrast to other probabilistic loss extensions such as $f$-DPO [62] and $f$-PO [67], which show a clear trade-off between win rate and diversity, BPO instances improve both win rate and entropy over DPO, as summarized in Figure 1. The proposed SBA instance achieves the best trade-off between win rate and diversity among all instances. We also apply the proposed loss to Llama-3-8B-Instruct [18] and achieve a 55.9% length-controlled win rate against GPT-4 Turbo on AlpacaEval2 [36]. To the best of our knowledge, this is state-of-the-art performance among Llama-3-8B backbone models.

## 2 Preliminaries

### 2.1 Alignment of large language models with human preference

We consider an autoregressive language model $\pi_\theta$ as a policy model that generates a response sequence $\mathbf{y} = [y^1, \ldots, y^L]$ conditioned on a prompt $\mathbf{x}$, with probability $\pi_\theta(\mathbf{y}|\mathbf{x}) = \prod_{k=1}^{L} \pi_\theta(y^k|\mathbf{x}, \mathbf{y}^{<k})$, where $L$ denotes the sequence length. Alignment methods commonly begin with a pre-trained large language model (LLM), followed by supervised fine-tuning (SFT) on a high-quality dataset. The resulting model is used as the reference model $\pi_{\text{ref}}$, and the policy model $\pi_\theta$ is initialized from $\pi_{\text{ref}}$.

**Reinforcement learning from human feedback (RLHF):** The alignment method RLHF [6, 47, 52, 74] consists of two phases following SFT. The first phase involves learning a reward model $r_\phi$. This phase requires access to an offline pairwise preference dataset, consisting of tuples $(\mathbf{x} :$ prompt, $\mathbf{y}_w :$ preferred response, $\mathbf{y}_l :$ dispreferred response$)$ drawn from the underlying preference data distribution $p_{\text{data}}(\mathbf{y}_w \succ \mathbf{y}_l|\mathbf{x})p_{\text{prompt}}(\mathbf{x})$. The learning objective of the reward model is

$$\mathcal{L}_{\text{reward}}(r_\phi; p_{\text{data}}) = -\mathbb{E}_{p_{\text{data}}(\mathbf{y}_w \succ \mathbf{y}_l|\mathbf{x})}[\log\sigma(r_\phi(\mathbf{x}, \mathbf{y}_w) - r_\phi(\mathbf{x}, \mathbf{y}_l))], \tag{1}$$

where $\sigma$ denotes the logistic function. The prompt $\mathbf{x}$ is drawn from $p_{\text{prompt}}(\mathbf{x})$, which we omit in the following formulations for notational simplicity. From the optimality condition of logistic regression, the optimal reward model $r_{\phi^*}$ that minimizes $\mathcal{L}_{\text{reward}}$ satisfies $p_{\text{data}}(\mathbf{y}_w \succ \mathbf{y}_l|\mathbf{x}) = \sigma(r_{\phi^*}(\mathbf{x}, \mathbf{y}_w) - r_{\phi^*}(\mathbf{x}, \mathbf{y}_l))$, also known as the Bradley–Terry model [9]. The next phase is RL fine-tuning based on proximal policy optimization (PPO) [50], formulated as

$$\mathcal{L}_{\text{RLHF}}(\pi_\theta; \pi_{\text{ref}}, r_{\phi^*}) = -\mathbb{E}_{\pi_\theta(\mathbf{y}|\mathbf{x})}[r_{\phi^*}(\mathbf{x}, \mathbf{y})] + \beta D_{\text{KL}}(\pi_\theta(\mathbf{y}|\mathbf{x})||\pi_{\text{ref}}(\mathbf{y}|\mathbf{x})). \tag{2}$$

Table 2: Examples of Bregman divergence instances defined by choices of $h$, and their corresponding BPO loss functions.

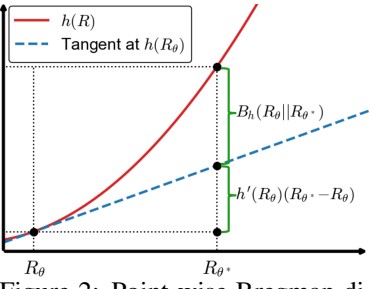

| Name | $h(R)$ | $\mathcal{L}^h_{\text{BPO}}(R_\theta; p_{\text{data}})$ |
|---|---|---|
| LR (DPO) | $\frac{R\log R - (1+R)\log(1+R)}{2}$ | $\mathbb{E}_{p_{\text{data}}}[\log(R_\theta + 1)]$ |
| KLIEP | $R\log R - R$ | $\mathbb{E}_{p_{\text{data}}}[R_\theta + \log R_\theta]$ |
| LSIF | $(R-1)^2$ | $\mathbb{E}_{p_{\text{data}}}[R_\theta^2 - \frac{2}{R_\theta}]$ |
| BA | $\frac{(R^{1+\lambda}-R)}{\lambda}$ | $\mathbb{E}_{p_{\text{data}}}[R_\theta^{\lambda+1} - \frac{\lambda+1}{\lambda}R_\theta^{-\lambda}]$ |
| SBA (Ours) | $\frac{(R^{1+\lambda}-R)}{s\lambda(\lambda+1)}$ | $\mathbb{E}_{p_{\text{data}}}\left[\frac{1}{s(\lambda+1)}R_\theta^{\lambda+1} - \frac{1}{s\lambda}R_\theta^{-\lambda}\right]$ |

Figure 2: Point-wise Bregman divergence defined by $h$, between the model $R_\theta$ and the target $R_{\theta^*}$.

$\mathcal{L}_{\text{RLHF}}$ consists of a reward maximization term and a $\beta$-weighted reverse KL regularization term that penalizes deviation of the policy from the reference model. Due to the complexity and instability of this multi-stage pipeline, direct preference optimization has become a strong alternative for alignment.

**Direct preference optimization (DPO):** The alignment method DPO [49] unifies the objectives Eqs. (1) and (2) into a single objective without the reward model. Eq. (2) has a closed-form solution:

$$\pi_{\theta^*}(\mathbf{y}|\mathbf{x}) = \arg\min_{\pi_\theta} \mathcal{L}_{\text{RLHF}} = \frac{1}{Z(\mathbf{x})}\pi_{\text{ref}}(\mathbf{y}|\mathbf{x})\exp\left(\frac{1}{\beta}r_{\phi^*}(\mathbf{x},\mathbf{y})\right), \quad (3)$$

where $Z(\mathbf{x}) = \sum_{\mathbf{y}}\pi_{\text{ref}}(\mathbf{y}|\mathbf{x})\exp\left(\frac{1}{\beta}r_{\phi^*}(\mathbf{x},\mathbf{y})\right)$ is the partition function. From Eq. (3), DPO defines a new reward model $r_{\text{DPO}}(\mathbf{x},\mathbf{y}) = \beta\log\frac{\pi_\theta(\mathbf{y}|\mathbf{x})}{\pi_{\text{ref}}(\mathbf{y}|\mathbf{x})} + \beta\log Z(\mathbf{x})$, which is parameterized by the policy and reference models. Substituting $r_{\text{DPO}}$ into $\mathcal{L}_{\text{reward}}(r_{\text{DPO}}; p_{\text{data}})$ yields the following objective:

$$\mathcal{L}_{\text{DPO}}(\pi_\theta; \pi_{\text{ref}}, p_{\text{data}}) = -\mathbb{E}_{p_{\text{data}}(\mathbf{y}_w \succ \mathbf{y}_l | \mathbf{x})}\left[\log\sigma\left(\beta\log\frac{\pi_\theta(\mathbf{y}_w|\mathbf{x})}{\pi_{\text{ref}}(\mathbf{y}_w|\mathbf{x})} - \beta\log\frac{\pi_\theta(\mathbf{y}_l|\mathbf{x})}{\pi_{\text{ref}}(\mathbf{y}_l|\mathbf{x})}\right)\right], \quad (4)$$

which enables the policy model to learn directly from a preference dataset in a supervised manner. The *optimal policy* $\pi_{\theta^*}$ that minimizes $\mathcal{L}_{\text{DPO}}$ matches the optimal solution in Eq. (3), and it satisfies:

$$p_{\text{data}}(\mathbf{y}_w \succ \mathbf{y}_l | \mathbf{x}) = \sigma\left(\beta\log\frac{\pi_{\theta^*}(\mathbf{y}_w|\mathbf{x})}{\pi_{\text{ref}}(\mathbf{y}_w|\mathbf{x})} - \beta\log\frac{\pi_{\theta^*}(\mathbf{y}_l|\mathbf{x})}{\pi_{\text{ref}}(\mathbf{y}_l|\mathbf{x})}\right). \quad (5)$$

Despite the theoretical guarantees, optimality is often not achieved in practice because of limited model capacity and imperfect optimization. Since the practical solution depends on the form of the objective function, generalizing the objective provides potential benefits.

## 2.2  Likelihood ratio estimation under Bregman divergence

Given two probability distributions $p_{\text{de}}(\mathbf{x})$ and $p_{\text{nu}}(\mathbf{x})$, likelihood ratio estimation aims to learn a ratio model $R_\theta(\mathbf{x})$ that approximates $R_{\text{data}}(\mathbf{x}) := \frac{p_{\text{nu}}(\mathbf{x})}{p_{\text{de}}(\mathbf{x})}$, based on i.i.d. samples from both distributions. Probabilistic classification via logistic regression [21] is the most commonly used approach. Traditional methods such as the Kullback-Leibler importance estimation procedure (KLIEP) [45, 54] and least-squares importance fitting (LSIF) [25, 20, 29] have also been widely used. These methods can be unified under the Bregman divergence [10] framework in [53], resulting in the following formulation:

$$D_h\big(R_{\text{data}}(\mathbf{x})||R_\theta(\mathbf{x})\big) = \int p_{\text{de}}(\mathbf{x})B_h\big(R_{\text{data}}(\mathbf{x})||R_\theta(\mathbf{x})\big)d\mathbf{x}$$

$$= \int p_{\text{de}}(\mathbf{x})\big(h\big(R_{\text{data}}(\mathbf{x})\big) - h(R_\theta(\mathbf{x})) - h'\big(R_\theta(\mathbf{x})\big)\big(R_{\text{data}}(\mathbf{x}) - R_\theta(\mathbf{x})\big)\big)d\mathbf{x}, \quad (6)$$

where $h$ denotes a strictly convex and twice continuously differentiable function with derivative function $h'$, and $B_h$ is the pointwise Bregman divergence which measures the error of the linear approximation as illustrated in Figure 2. Specific instances are summarized in Table 2, showing that previous likelihood ratio estimation methods differ only by the choice of $h$. Among various instances, Basu's power (BA) divergence [7], defined for $\lambda > -1$, smoothly interpolates between KLIEP (at $\lambda = 0$) and LSIF (at $\lambda = 1$). Inspired by the success of generalized likelihood ratio estimation in recent generative modeling studies [38, 30, 32], we apply this extension to generalize the DPO loss.

# 3 Methods

This section introduces the proposed *Bregman Preference Optimization* (BPO). Section 3.1 reformulates the DPO objective as a likelihood ratio estimation problem. Section 3.2 extends the DPO objective via Bregman divergence under the likelihood ratio estimation perspective. Section 3.3 analyzes instances of BPO and introduces the scaled Basu's power (SBA) divergence within this framework. Finally, Section 3.4 discusses the applicability of BPO to other DPO variants in an orthogonal manner.

## 3.1 Preference optimization as likelihood ratio estimation

In contrast to prior works [27, 67] that characterize the optimal policy using a learned reward model $r_{\phi*}$ and a partition function $Z(\mathbf{x})$ as in Eq. (3), we aim to characterize the optimal policy without such modeling complexity.

**Proposition 1.** *Let the optimal policy* $\pi_{\theta*} := \arg\min_{\pi_\theta} \mathcal{L}_{DPO}(\pi_\theta; \pi_{ref}, p_{data})$, *the following holds:*

$$\frac{\pi_{\theta*}(\mathbf{y}_w|\mathbf{x})}{\pi_{\theta*}(\mathbf{y}_l|\mathbf{x})} = \frac{\pi_{ref}(\mathbf{y}_w|\mathbf{x})}{\pi_{ref}(\mathbf{y}_l|\mathbf{x})} \times \left(\frac{p_{data}(\mathbf{y}_w \succ \mathbf{y}_l|\mathbf{x})}{p_{data}(\mathbf{y}_w \prec \mathbf{y}_l|\mathbf{x})}\right)^{1/\beta}. \tag{7}$$

Please refer to Appendix A.1 for the proof. Eq. (7) shows that the likelihood ratio of optimal policy $\pi_{\theta*}$ can be specified solely using the reference model $\pi_{\text{ref}}$ and the preference data distribution $p_{\text{data}}$. The ratio $\frac{\pi_{\theta*}(\mathbf{y}_w|\mathbf{x})}{\pi_{\theta*}(\mathbf{y}_l|\mathbf{x})}$ is equivalent to the *concrete score* [41, 38] (up to a constant), and the concrete score satisfies the completeness property [39, 41]. This means that the concrete score can uniquely identify the distribution $\pi_{\theta*}$. Therefore, the matching $\frac{\pi_\theta(\mathbf{y}_w|\mathbf{x})}{\pi_\theta(\mathbf{y}_l|\mathbf{x})}$ to $\frac{\pi_{\theta*}(\mathbf{y}_w|\mathbf{x})}{\pi_{\theta*}(\mathbf{y}_l|\mathbf{x})}$ is sufficient for $\pi_\theta$ to recover $\pi_{\theta*}$, providing a theoretical justification for ratio matching. To build a data-driven estimator for this matching, we rearrange Eq. (7), as detailed in Eq. (15) to Eq. (19), and reformulate preference optimization as a matching problem between data ratio $R_{\text{data}}$ and model ratio $R_\theta$, defined as:

$$R_{\text{data}}(\mathbf{x}, \mathbf{y}_w, \mathbf{y}_l) := \frac{p_{\text{data}}(\mathbf{y}_w \prec \mathbf{y}_l|\mathbf{x})}{p_{\text{data}}(\mathbf{y}_w \succ \mathbf{y}_l|\mathbf{x})}, \qquad R_\theta(\mathbf{x}, \mathbf{y}_w, \mathbf{y}_l) := \left[\frac{\pi_\theta(\mathbf{y}_l|\mathbf{x})\pi_{\text{ref}}(\mathbf{y}_w|\mathbf{x})}{\pi_\theta(\mathbf{y}_w|\mathbf{x})\pi_{\text{ref}}(\mathbf{y}_l|\mathbf{x})}\right]^\beta. \tag{8}$$

## 3.2 Bregman preference optimization via ratio matching

As discussed in Section 3.1, preference optimization can be formulated as a ratio matching between $R_{\text{data}}$ and $R_\theta$. We define the loss using the Bregman ratio matching framework in Section 2.2 as:

$$D_h(R_{\text{data}}||R_\theta) = \mathbb{E}_{p_{\text{data}}(\mathbf{y}_w \succ \mathbf{y}_l|\mathbf{x})}[h(R_{\text{data}}) - h(R_\theta) - h'(R_\theta)(R_{\text{data}} - R_\theta)], \tag{9}$$

where $R_{\text{data}}$ and $R_\theta$ denote the ratios evaluated at $(\mathbf{x}, \mathbf{y}_w, \mathbf{y}_l)$ on the right-hand side, $h$ denotes a strictly convex and twice continuously differentiable function with derivative function $h'$.

**Theorem 2.** *(Optimality) Under sufficient model capacity,* $\arg\min_{\pi_\theta} D_h(R_{data}||R_\theta) = \pi_{\theta*}$.

Please see Appendix A.2 for the proof. Theorem 2 implies that the objective $D_h(R_{\text{data}}||R_\theta)$ guarantees target optimality under a valid $h$, but it is intractable since evaluating $R_{\text{data}}$ at a given point $(\mathbf{x}, \mathbf{y}_w, \mathbf{y}_l)$ is infeasible. As preference optimization provides only samples from $p_{\text{data}}(\mathbf{y}_w \succ \mathbf{y}_l \mid \mathbf{x})$ without direct access to the distribution, we propose a tractable alternative inspired by implicit score matching [26]:

$$\mathcal{L}_{\text{BPO}}^h(R_\theta; p_{\text{data}}) := \mathbb{E}_{p_{\text{data}}(\mathbf{y}_w \succ \mathbf{y}_l|\mathbf{x})}\left[h'(R_\theta)R_\theta - h(R_\theta) - h'\left(R_\theta^{-1}\right)\right], \tag{10}$$

and the equivalence is guaranteed by the following theorem.

**Theorem 3.** $\mathcal{L}_{BPO}^h(R_\theta; p_{data}) = D_h(R_{data}||R_\theta) + C$, *where $C$ is constant with respect to $\theta$.*

The proof is provided in Appendix A.3. The intractable term $R_{\text{data}}$ does not appear in $\mathcal{L}_{\text{BPO}}^h$, and the statistical information of $R_{\text{data}}$ is captured entirely via samples from $p_{\text{data}}$. As the value of function $R_\theta$ at the point $(\mathbf{x}, \mathbf{y}_w, \mathbf{y}_l)$ computed through the feed-forward passes of $\pi_\theta$ and $\pi_{\text{ref}}$, $R_\theta$ is appropriate to retain it inside the expectation. To analyze the learning dynamics of $\mathcal{L}_{\text{BPO}}^h(R_\theta; p_{\text{data}})$, we provide the following gradient analysis:

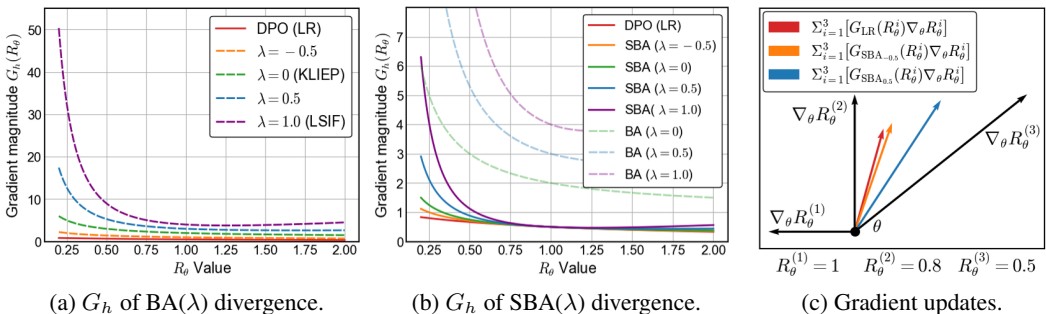

(a) $G_h$ of BA($\lambda$) divergence.  (b) $G_h$ of SBA($\lambda$) divergence.  (c) Gradient updates.

Figure 3: Gradient magnitude and direction analysis across different Bregman divergences.

**Proposition 4.** *(Gradient Analysis) Let the gradient of the BPO objective be expressed as:*

$$\nabla_\theta \mathcal{L}_{BPO}^h(R_\theta; p_{data}) = \mathbb{E}_{p_{data}(\mathbf{y}_w \succ \mathbf{y}_l|\mathbf{x})}\left[G_h(R_\theta)\nabla_\theta R_\theta\right], \tag{11}$$

*where we define $G_h(R_\theta)$ as the magnitude of gradient. If $h$ is strictly convex and twice continuously differentiable, then $G_h(R_\theta) > 0$ for all $R_\theta$.*

See Appendix A.4 for the proof. Eq. (11) shows that the gradient direction at a point $(\mathbf{x}, \mathbf{y}_w, \mathbf{y}_l)$ depends only on $\nabla_\theta R_\theta$, regardless of the choice of $h$. In contrast, $h$ controls the point-wise gradient magnitude $G_h(R_\theta)$, which determines the relative weighting of each sample during optimization. The weighting can influence the gradient direction aggregated over a mini-batch as shown in Figure 3c. Proposition 4 further explains that gradient descent updates decrease the value of $R_\theta$ at the observed point $(\mathbf{x}, \mathbf{y}_w, \mathbf{y}_l) \sim p_{data}$. This is intuitive since the observed pair is sampled from the distribution in the denominator of the target ratio $R_{data} = \frac{p_{data}(\mathbf{y}_w \prec \mathbf{y}_l|\mathbf{x})}{p_{data}(\mathbf{y}_w \succ \mathbf{y}_l|\mathbf{x})}$. Figures 3a and 3b show that $G_h(R_\theta)$ remains positive across various choices of $h$. Although Theorem 2 shows that the theoretical optimality holds regardless of the choice of the function $h$, the gradient magnitude $G_h(R_\theta)$ varies with $h$, making the choice of $h$ important due to potential sub-optimality in practical optimization.

### 3.3 Instances of Bregman preference optimization

Table 2 summarizes well-known instances of Bregman divergences unified in [53] and their corresponding BPO objectives $\mathcal{L}_{BPO}^h$ determined by the choice of $h$.

**BPO recovers DPO as a special case:** The proposed $\mathcal{L}_{BPO}^h$ recovers the original DPO objective $\mathcal{L}_{DPO}$ when $h$ corresponds to the logistic regression, denoted as $\mathcal{L}_{BPO}^{LR}$ (See Appendix A.5 for more details):

$$\mathcal{L}_{BPO}^{LR}(R_\theta; p_{data}) = \mathbb{E}_{p_{data}(\mathbf{y}_w \succ \mathbf{y}_l|\mathbf{x})}\left[\log(1 + R_\theta)\right]$$

$$= \mathbb{E}_{p_{data}(\mathbf{y}_w \succ \mathbf{y}_l|\mathbf{x})}\left[\log\left(1 + \left[\frac{\pi_\theta(\mathbf{y}_l|\mathbf{x})\pi_{ref}(\mathbf{y}_w|\mathbf{x})}{\pi_\theta(\mathbf{y}_w|\mathbf{x})\pi_{ref}(\mathbf{y}_l|\mathbf{x})}\right]^\beta\right)\right]$$

$$= \mathbb{E}_{p_{data}(\mathbf{y}_w \succ \mathbf{y}_l|\mathbf{x})}\left[-\log\sigma\left(\beta\log\frac{\pi_\theta(\mathbf{y}_w|\mathbf{x})}{\pi_{ref}(\mathbf{y}_w|\mathbf{x})} - \beta\log\frac{\pi_\theta(\mathbf{y}_l|\mathbf{x})}{\pi_{ref}(\mathbf{y}_l|\mathbf{x})}\right)\right] = \mathcal{L}_{DPO}(\pi_\theta; \pi_{ref}, p_{data}).$$

**KLIEP and LSIF under Basu's power divergence (BA):** The instances, $\mathcal{L}_{BPO}^{KLIEP}$ and $\mathcal{L}_{BPO}^{LSIF}$, are unified under $\mathcal{L}_{BPO}^{BA_\lambda}$, where the gradient magnitude is given by:

$$G_{BA_\lambda}(R_\theta) = (\lambda + 1)(R_\theta^\lambda + R_\theta^{-\lambda - 1}), \tag{12}$$

as shown in Figure 3a for various values of $\lambda$. While the term $(R_\theta^\lambda + R_\theta^{-\lambda-1})$ provides meaningful control over the optimization behavior with respect to $R_\theta$ via the hyperparameter $\lambda$, the coefficient $(\lambda + 1)$ unnecessarily scales up the gradient magnitude without introducing any $\theta$-dependent learning signal. Compared to the gradient magnitude of DPO, given by $G_{LR}(R_\theta) = \frac{1}{1+R_\theta}$, the gradient magnitude of $G_{BA_\lambda}(R_\theta)$ increases significantly as $\lambda$ becomes larger. When the gradient scale varies, training requires careful adjustment of the learning rate, batch size, and optimizer. Since preference optimization is sensitive to hyperparameters, improper tuning can severely degrade performance. To mitigate this issue, we propose a simple scaled version of BA to match DPO's gradient scale.

**Scaled Basu's power divergence (SBA):** We propose $h(R) = \frac{R^{1+\lambda} - R}{s\lambda(\lambda+1)}$, which corresponds to scaling the $h$-function of BA by a factor of $s(\lambda+1)$, where $s$ is a scaling constant. This scaling directly affects the gradient magnitude as follows:

$$G_{\text{SBA}_\lambda}(R_\theta) = (R_\theta^\lambda + R_\theta^{-\lambda-1})/s. \tag{13}$$

$G_{\text{SBA}_\lambda}(R_\theta)$ eliminates the unnecessary $\lambda$-dependent amplification, resulting in a more reasonable gradient scale in Figure 3b. Since preference optimization typically initializes $\pi_\theta$ as $\pi_{\text{ref}}$, the value of $R_\theta$ is 1 for all input points $(\mathbf{x}, \mathbf{y}_w, \mathbf{y}_l)$ at the start of training. By setting $s = 4$, the gradient magnitude of $G_{\text{SBA}_\lambda}$ matches that of $G_{\text{LR}}$ at initialization ($R_\theta = 1$). The hyperparameter $\lambda$ allows the model to control whether to prioritize updates for more confident (e.g., $R_\theta \ll 1$) or less confident (e.g., $R_\theta \approx 1$) samples. Recent studies [64, 31] have shown that DPO is sensitive to data quality and that applying confidence-based adjustments to individual samples can be heuristically effective. SBA controls the sensitivity to confident samples by tuning $\lambda$, while preserving theoretical optimality.

### 3.4 Compatibility with other DPO extensions

We have discussed how the optimal policy defined by DPO can be approximated by the policy model. Our generalization can also be applied orthogonally to DPO loss variants that define different optimal policies. We consider $f$-DPO [62] as an example by defining a *model ratio* as:

$$R_\theta^{f\text{-DPO}}(\mathbf{x}, \mathbf{y}_w, \mathbf{y}_l) := \exp\left[ -\beta f'\left( \frac{\pi_\theta(\mathbf{y}_w|\mathbf{x})}{\pi_{\text{ref}}(\mathbf{y}_w|\mathbf{x})} \right) + \beta f'\left( \frac{\pi_\theta(\mathbf{y}_l|\mathbf{x})}{\pi_{\text{ref}}(\mathbf{y}_l|\mathbf{x})} \right) \right], \tag{14}$$

where $f : \mathbb{R}^+ \to \mathbb{R}$ denotes a convex function satisfying $f(1) = 0$. Substituting $R_\theta^{f\text{-DPO}}$ and the $h$ function corresponding to logistic regression into BPO; $\mathcal{L}_{\text{BPO}}^{\text{LR}}(R_\theta^{f\text{-DPO}}; p_{\text{data}})$ recovers the original objective of $f$-DPO [62] (See Appendix A.6 for details). BPO enables constructing new objectives by varying the choice of $h$ based on the model ratio $R_\theta^{f\text{-DPO}}$, which results in $\mathcal{L}_{\text{BPO}}^h(R_\theta^{f\text{-DPO}}; p_{\text{data}})$. Similarly, existing logistic regression-based DPO variants can also be generalized by BPO.

## 4 Experiments

This section presents the empirical performance of the proposed BPO compared to prior preference optimization methods. Section 4.1 compares instances of BPO with other probabilistic loss extensions. Section 4.2 compares BPO to the state-of-the-art DPO loss variants on popular LLM benchmarks.

### 4.1 Comparison with probabilistic DPO loss extensions

We compare BPO with baseline methods for probabilistic DPO loss extensions, and analyze the effect of different choices of the function $h$ within the BPO framework.

**Task & experimental setup:** The experiments are conducted for single-turn dialogue generation using the Anthropic helpful and harmless (HH) dataset [6], and summarization using the Reddit TL;DR dataset [59]. While the experimental setups vary across the different baseline papers, we faithfully follow the setup provided in the original DPO paper [49] for a fair comparison. For dialogue generation, we use Pythia-2.8B [8] as the pre-trained LLM and perform SFT on the preferred subset of the HH dataset. For summarization, we use a publicly available SFT model [11] based on GPT-J [61]. All comparisons are conducted on the same SFT model with identical training hyperparameters ($\beta$, learning rate, batch size). See Appendix C for details.

**Baselines:** We compare with representative baselines, including instances of $f$-DPO [62] and $f$-PO [67]. For $f$-DPO, we consider forward KL (FKL), Jensen-Shannon (JS), $\alpha$-divergence with $\alpha = 0.1, 0.3, 0.5$, and 0.7, selecting $\alpha = 0.1$ based on the best win rate, and additionally include $\chi^2$-divergence [24]. For $f$-PO, we consider Jeffrey, JS, reverse KL (RKL) [27], and $\alpha$-divergence with $\alpha = 0.1$, as suggested in the original paper. To ensure a fair computational comparison, we adopt the pairwise approximation proposed in $f$-PO.

**Evaluation metrics:** Both fidelity and diversity of the generated samples are important from a probability matching perspective. We evaluated the models using test datasets. To measure fidelity, we use the win rate against both the preferred responses and the SFT model responses, as judged by GPT-4. To assess diversity, we use predictive entropy [70, 62], self-BLEU [73], and distinct-1 [34].

Table 3: Comparison of dialogue generation performance on Anthropic-HH, using the same SFT model based on Pythia-2.8B. Each value is colored red if it is better than DPO, and blue if it performs worse. The best result for each metric is shown in **bold**, and the second and third best are underlined.

| Type | Loss | Win rate (%) ↑ | | Diversity | | |
|---|---|---|---|---|---|---|
| | | vs Preferred | vs SFT | Entropy ↑ | BLEU ↓ | Distinct-1 ↑ |
| - | SFT | 33.5 | - | 3.508 | 0.096 | 0.375 |
| All | DPO | 48.5 | 71.5 | 2.801 | 0.145 | 0.336 |
| *f*-DPO [62] | FKL | $34.5^{-28.9\%}$ | $55.5^{-22.4\%}$ | $\mathbf{3.354}^{+19.7\%}$ | $\mathbf{0.088}^{+39.3\%}$ | $\underline{0.338}^{+0.7\%}$ |
| | $\alpha$-div. | $47.5^{-2.1\%}$ | $64.0^{-10.5\%}$ | $\underline{3.125}^{+11.6\%}$ | $\underline{0.123}^{+14.9\%}$ | $0.332^{-1.2\%}$ |
| | JS | $52.0^{+7.2\%}$ | $70.0^{-2.1\%}$ | $2.723^{-2.8\%}$ | $0.139^{+3.8\%}$ | $0.299^{-10.9\%}$ |
| | $\chi^2$ [24] | $\underline{53.5}^{+10.3\%}$ | $72.0^{+0.7\%}$ | $2.369^{-15.4\%}$ | $0.134^{+7.3\%}$ | $0.279^{-16.8\%}$ |
| *f*-PO [67] | Jeffrey | $51.5^{+6.2\%}$ | $68.0^{-4.9\%}$ | $2.436^{-13.0\%}$ | $0.166^{-14.7\%}$ | $0.271^{-19.1\%}$ |
| | JS | $\underline{54.5}^{+12.4\%}$ | $\underline{76.0}^{+6.3\%}$ | $2.531^{-9.6\%}$ | $0.154^{-6.1\%}$ | $0.283^{-15.8\%}$ |
| | $\alpha$-div. | $52.0^{+7.2\%}$ | $\underline{75.5}^{+5.6\%}$ | $2.298^{-18.0\%}$ | $0.168^{-15.9\%}$ | $0.290^{-13.6\%}$ |
| | RKL [27] | $49.5^{+2.1\%}$ | $70.5^{-1.4\%}$ | $2.321^{-17.1\%}$ | $0.182^{-25.8\%}$ | $0.268^{-20.2\%}$ |
| BPO | BA | $51.0^{+5.2\%}$ | $75.5^{+5.6\%}$ | $2.803^{+0.1\%}$ | $0.141^{+2.6\%}$ | $0.320^{-4.5\%}$ |
| | KELIP | $48.5^{+0.0\%}$ | $71.5^{+0.0\%}$ | $2.901^{+3.6\%}$ | $0.151^{-4.1\%}$ | $0.332^{-1.0\%}$ |
| | LSIF | $50.5^{+4.1\%}$ | $72.5^{+1.4\%}$ | $2.908^{+3.8\%}$ | $0.139^{+4.1\%}$ | $0.321^{-4.5\%}$ |
| | SBA | $\mathbf{57.0}^{+17.5\%}$ | $\mathbf{77.0}^{+7.7\%}$ | $\underline{3.010}^{+7.5\%}$ | $\underline{0.132}^{+9.0\%}$ | $\mathbf{0.340}^{+1.4\%}$ |

Table 4: The performance of $\mathcal{L}_{\text{BPO}}^h(R_\theta^{f\text{-DPO}}; p_{\text{data}})$, which extends *f*-DPO [62] orthogonally using our BPO framework.

| $f(\cdot)$ | $h(\cdot)$ | Win rate (%) ↑ | | Diversity | | |
|---|---|---|---|---|---|---|
| | | vs Preferred | vs SFT | Entropy ↑ | BLEU ↓ | Distinct-1 ↑ |
| FKL | LR | 34.5 | 55.5 | **3.354** | 0.088 | 0.338 |
| | SBA | **39.5** | **60.0** | 3.237 | **0.081** | **0.340** |
| JS | LR | 52.0 | 70.0 | 2.723 | 0.139 | 0.299 |
| | SBA | **55.0** | **73.5** | **2.856** | **0.127** | **0.307** |

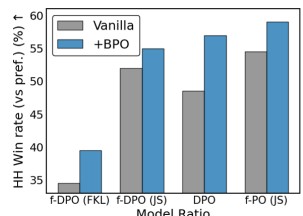

Figure 4: Orthogonal improvements with BPO (SBA).

#### 4.1.1 Dialogue generation task

**Comparison to baselines:** Figure 1 shows the trade-off between win rate and entropy for each instance, and Table 3 summarizes the results across all metrics. *f*-DPO recovers DPO when RKL is used as the regularizer. As the $\alpha$-divergence shifts from RKL to FKL, the win rate decreases while diversity improves, reflecting the mode-covering nature of FKL. $\chi^2$ improves win rate, as observed in $\chi^2$-PO [25]. *f*-PO recovers DPO with FKL. Other divergences, such as Jeffrey, JS, $\alpha$-divergence, and RKL, have weaker mode coverage than FKL, resulting in worse diversity across all metrics.

All BPO instances achieve win rates at least as high as those of DPO against both preferred responses and SFT responses, while also achieving better entropy. *f*-DPO and *f*-PO exhibit substantial degradations in some cases, such as a 28.9% win rate drop in *f*-DPO (FKL) and a 25.8% BLEU drop in *f*-PO (RKL). In contrast, instances of BPO show no degradation greater than 5% on any metric. Notably, SBA is the only instance that consistently outperforms DPO on all metrics, achieving the highest win rate compared to all other losses.

**Orthogonal utilization with other methods:** As discussed in Section 3.4, existing loss functions that can be expressed as logistic regression can be orthogonally extended using BPO. Table 4 reports the performance of BPO applied to two instances of *f*-DPO (FKL and JS), showing improvements in 9 out of 10 metrics. As shown in Appendix B.2, *f*-PO with pairwise datasets can also be viewed as logistic regression for some $f$, and applying SBA to *f*-PO (JS) results in further performance gains. Figure 4 summarizes the performance improvements achieved by applying SBA to each model ratio.

**Ablation studies on $\lambda$ in SBA:** Figure 5 shows ablation studies on the effect of $\lambda$ in the proposed SBA, an instance of the BPO framework. When $\lambda = -0.5$, the gradient behavior closely resembles that of DPO, as shown in Figure 3b, with both win rate and entropy showing similar trends in Figures 5a and 5b. As $\lambda$ increases, both metrics improve up to a certain point, in contrast to the

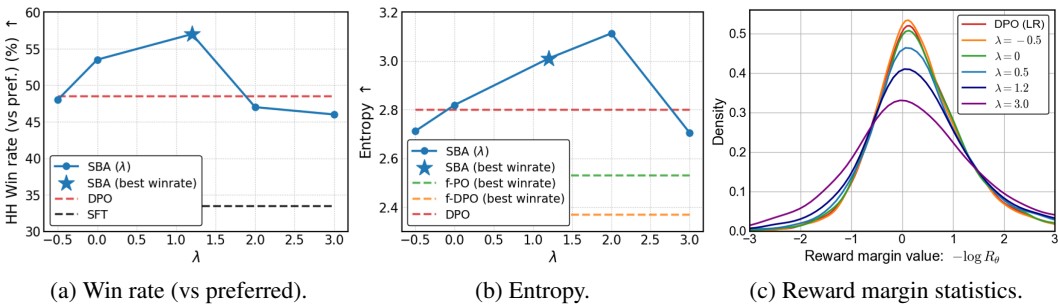

(a) Win rate (vs preferred).  (b) Entropy.  (c) Reward margin statistics.

Figure 5: Ablation studies on the effect of $\lambda$ in SBA, using Anthropic-HH with Pythia-2.8B.

Table 5: Performance on TL;DR summarization based on GPT-J with 0.25 temperature.

| Type | Win rate (%) ↑ | | Diversity | |
|---|---|---|---|---|
| | vs Preferred | vs SFT | BLEU ↓ | Entropy ↑ |
| DPO | 47.0 | 63.0 | 0.597 | 0.276 |
| $f$-DPO | 46.0 | 60.5 | 0.596 | 0.258 |
| $f$-PO | 52.5 | 70.5 | 0.571 | 0.302 |
| BPO | **61.0** | **71.0** | **0.565** | **0.318** |

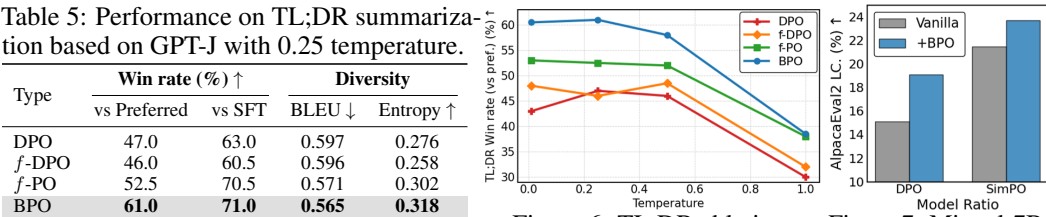

Figure 6: TL;DR ablation.   Figure 7: Mistral-7B.

instances of $f$-PO and $f$-DPO that exhibit a clear trade-off between win rate and entropy. Larger values of $\lambda$ lead the model to focus more on confident samples (where $R_\theta$ deviates further from 1), resulting in a wider spread in reward margin ($-\log R_\theta$) statistics, as shown in Figure 5c. This shift in the statistics of reward margins appears to be an underlying factor contributing to the performance improvements.

#### 4.1.2 Summarization task

For the TL;DR summarization task, we compare the best win rate instances of each framework reported in Table 3, where $f$-DPO uses $\chi^2$, $f$-PO uses JS, and BPO uses SBA. Figure 6 presents the win rate against preferred responses across different sampling temperatures, showing a trend consistent with prior studies that lower temperatures lead to better performance. BPO consistently outperforms the other instances at all temperatures. Table 5 shows additional metrics at temperature 0.25, where BPO shows the best performance across all metrics.

### 4.2 Comparison with general DPO loss variants

This section extends the analysis to a larger model and evaluates its performance on external benchmarks beyond the training domain, comparing it with general DPO variants.

**Experimental setup:** We conduct experiments using Mistral-7B-Base [28], Llama-3-8B-Base, and Llama-3-8B-Instruct [18] backbone models, based on the UltraFeedback dataset [15]. For Mistral-7B-Base, we use publicly available SFT models from Zephyr [58, 57]. We extend DPO with our BPO loss based on the Zephyr setup and extend SimPO with our BPO loss following the SimPO [43] setup. For Llama-3-8B-Base and Llama-3-8B-Instruct, we use the same SFT models [19] as in the SimPO setup, along with the adapted version of the UltraFeedback dataset [42]. See Appendix C for details.

**Evaluation benchmark:** We evaluate the response capability across a wide range of queries using the most popular open-ended instruction-following benchmarks. AlpacaEval2 [36] measures the win rate on 805 examples, using GPT-4 Turbo as both the judge and the opponent. The evaluation covers both raw and length-controlled responses. Arena-Hard [35] measures the win rate on 500 queries, using GPT-4 Turbo as the judge and GPT-4-0314 as the opponent. All evaluations follow the library configurations and decoding parameters used in SimPO to ensure fair comparisons.

**Results:** Table 6 presents the main results obtained with various backbone models. Baseline results are taken from SimPO [43]. Most DPO variants exhibit even worse performance than the standard DPO. BPO can also be applied on top of SimPO, as discussed in Section 3.4, and it achieves consistent performance gains over SimPO, except in one out of nine cases. To the best of our knowledge, BPO

Table 6: Performance on AlpacaEval2 and Arena-Hard with various backbone models. **LC** (%) indicates the length-controlled win rate, and **WR** (%) denotes the raw win rate. The baseline results are from SimPO [43].

| Method | Mistral-7B-Base | | | Llama-3-8B-Base | | | Llama-3-8B-Instruct | | |
| | AlpacaEval | | Arena-Hard | AlpacaEval | | Arena-Hard | AlpacaEval | | Arena-Hard |
| | LC | WR | WR | LC | WR | WR | LC | WR | WR |
|---|---|---|---|---|---|---|---|---|---|
| SFT | 8.4 | 6.2 | 1.3 | 6.2 | 4.6 | 3.3 | 26.0 | 25.3 | 22.3 |
| RRHF [69] | 11.6 | 10.2 | 5.8 | 12.1 | 10.1 | 6.3 | 37.9 | 31.6 | 28.8 |
| SLiC-HF [72] | 10.9 | 8.9 | 7.3 | 12.3 | 13.7 | 6.0 | 33.9 | 32.5 | 29.3 |
| DPO [49] | 15.1 | 12.5 | 10.4 | 18.2 | 15.5 | 15.9 | 48.2 | 47.5 | 35.2 |
| IPO [5] | 11.8 | 9.4 | 7.5 | 14.4 | 14.2 | 17.8 | 46.8 | 42.4 | 36.6 |
| CPO [66] | 9.8 | 8.9 | 6.9 | 10.8 | 8.1 | 5.8 | 34.1 | 36.4 | 30.9 |
| KTO [16] | 13.1 | 9.1 | 5.6 | 14.2 | 12.4 | 12.5 | 34.1 | 32.1 | 27.3 |
| ORPO [23] | 14.7 | 12.2 | 7.0 | 12.2 | 10.6 | 10.8 | 38.1 | 33.8 | 28.2 |
| R-DPO [48] | 17.4 | 12.8 | 8.0 | 17.6 | 14.4 | 17.2 | 48.0 | 45.8 | 35.1 |
| SimPO [43] | 21.5 | 20.8 | 16.6 | 22.0 | **20.3** | 23.4 | 53.7 | 47.5 | 36.5 |
| BPO | **23.7** | **20.9** | **16.9** | **22.5** | 18.7 | **31.7** | **55.9** | **51.5** | **38.0** |

achieves state-of-the-art **LC** performance among preference-optimized models based on Llama-3-8B-Instruct. Figure 7 shows results with Mistral-7B-Base. Specifically, BPO improves **LC** from 15.1% to 19.1% when built on DPO, and from 21.5% to 23.7% when built on SimPO. These results demonstrate that BPO generalizes well to external benchmarks beyond the training domain with the 7–8B LLM scale, showing consistent improvements across different backbones and model ratio formulations, including DPO and SimPO.

## 5 Related work

A series of studies have proposed variants of the DPO loss. SLiC [72] and IPO [5] replace the logistic regression in DPO with hinge and squared losses, respectively, while GPO [55] unifies these variants within a binary classification framework. KTO [16] generalizes binary preference comparisons to multiwise settings. SPPO [65] formulates preference optimization as a two-player game between policies, framing the optimal policy as a Nash equilibrium and offering a game-theoretic interpretation of DPO. SimPO [43] and ORPO [23] proposed reference-free preference optimization, enabling more efficient and offline-compatible implementations. TDPO [70] applies the preference loss at the token level, enabling fine-grained reward assignment and improving alignment with human preferences. $\beta$-DPO [64] dynamically adjusts the regularization parameter $\beta$ based on the reward margin between preference pairs, improving training stability and reducing sensitivity. While these studies explored complementary directions to improve the flexibility, efficiency, and robustness of preference optimization, they offer limited probabilistic interpretation and rely primarily on heuristics.

**DPO loss generalization in distribution matching perspective:** $f$-DPO [62] generalizes the reverse KL regularization in $\mathcal{L}_{\text{RLHF}}$ (see Eq. (2)) to the broader class of $f$-divergences [14, 37], and deriving direct optimization objectives for a subset of $f$-divergences that satisfy specific conditions. $\chi^2$-PO [24] extends the direct optimization objective for $\chi^2$ divergence regularization within the $f$-DPO framework. MPO [2] generalizes the regularization under the Bregman divergence, particularly for meta-learning. However, modifying the regularization alone is potentially insufficient to determine the overall behavior of the loss function and naturally leads to a different optimal policy, as we discussed in Appendix B.1. EXO [27] reformulates the entire DPO objective as a distribution matching problem, showing that $\mathcal{L}_{\text{DPO}}$ in Eq. (4) reduces to the forward KL divergence between the policy model $\pi_\theta(\mathbf{y}|\mathbf{x})$ and the optimal policy $\pi_{\theta*}(\mathbf{y}|\mathbf{x})$ in Eq. (3). EXO further proposes a reverse KL loss, and $f$-PO [67] subsequently extends this formulation to $f$-divergences. In contrast to $f$-DPO, which yields a different optimal policy, $f$-PO is analogous to extensions of traditional generative modeling [46, 51] that directly match a policy to a target optimal policy. However, the $f$-PO loss requires a learned reward model $r_{\phi*}$ and Monte Carlo estimation of partition functions to compute the probability of the optimal policy $\pi_{\theta*}$. Simplifying $f$-PO for a paired preference dataset results in a different notion of target optimality as we discussed in Appendix B.2. As summarized in Table 1,

the proposed BPO is distinguished from existing probabilistic loss extensions by maintaining both optimality and simplicity, while allowing flexible choices of optimization objectives.

**Generative modeling and likelihood ratio estimation:** The likelihood ratio has played a central role in research on generative modeling. For continuous random variables, noise contrastive estimation (NCE) [21] estimates the ratio between a known noise distribution and the target distribution to approximate the target density. This ratio identifies the target density and also serves as a learning signal for neural samplers, inspiring subsequent work on GANs [17]. The noise distribution in NCE has been replaced by the model distributions of GANs [4, 12], VAEs [40, 3], and diffusion models [30, 44], and has recently been employed as a refinement technique in generative modeling. For discrete variables, concrete score matching (CSM) [41] estimates the ratio of probability masses between different states of a probability distribution. CSM has served as a foundation for the development of discrete diffusion modeling [38, 71], and this paper adopts the concept of CSM to autoregressive language models. Another concurrent work has utilized CSM for knowledge distillation in autoregressive language models [33].

## 6 Conclusion

We have introduced Bregman preference optimization (BPO), a generalized preference optimization objective based on Bregman divergence, formulated from the perspective of likelihood ratio estimation. BPO uniquely retains both optimality and simplicity among existing probabilistic loss extensions. Our gradient analysis further reveals that the optimization behavior depends on the choice of $h$, which determines the prioritization of samples, and we propose the gradient scaling method for BPO instances to facilitate training. Experimental results show that BPO consistently outperforms existing DPO variants across various scenarios, while being orthogonally applicable to other DPO variants.

In addition, recent advances in discrete diffusion models benefit from the likelihood ratio perspective [38]. We demonstrate that a similar extension is also effective for preference optimization using autoregressive language models. Although the types of probabilistic models and the datasets differ (i.e., unconditional generation vs. preference optimization), our results suggest a potential link through a shared probabilistic foundation toward a unified understanding of generative models.

**Limitations and broader impact** Despite promising experimental results on autoregressive language models, extending BPO to preference optimization of multi-modal LLMs [63] or diffusion models [60] remains an open direction for future work. As with other LLM research, advances in LLMs may improve user assistance in various tasks, but they also pose potential concerns about ethics and bias.

## Acknowledgment

This work was supported by the IITP (Institute of Information & Communications Technology Planning & Evaluation)-ITRC (Information Technology Research Center) grant funded by the Korea government (Ministry of Science and ICT) (IITP-2025-RS-2024-00437268).

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

# A Proofs and derivations

In this section, we provide detailed proofs for the theorems presented in the main text. Throughout this paper, we have the following assumption:

**Assumption.** $p_{data}(\mathbf{y}_w \succ \mathbf{y}_l \mid \mathbf{x}), \pi_{ref}(\mathbf{y} \mid \mathbf{x}), \pi_{\theta^*}(\mathbf{y} \mid \mathbf{x}) > 0$ *for all* $\mathbf{y}_w, \mathbf{y}_l, \mathbf{y}, \mathbf{x}$.

**Assumption.** *Function $h$ is strictly convex and twice continuously differentiable.*

These non-negative assumptions ensure that our target quantities, including likelihood ratios, are well-defined. The non-negative assumption for $p_{\text{data}}$ is naturally aligned with the Bradley-Terry model, which utilizes a sigmoid function to model pairwise probabilities. Given that this model inherently assumes a non-zero probability for each possible data pair, our assumption is a direct adaptation of this framework. We also assume that $\pi_{\text{ref}}$ remains strictly positive across its domain. This is reasonable, as we often construct $\mathbf{y}_w$ and $\mathbf{y}_l$ as being sampled from the reference policy distribution, where each possible outcome has a non-zero likelihood. Finally, the optimal policy $\pi_{\theta^*}$ is generally formulated as a reference policy with an exponential reward term, which implies that $\pi_{\theta^*}$ is non-zero whenever $\pi_{\text{ref}}$ is non-zero, making this assumption a natural extension of the positive support for the reference distribution.

According to the definition of Bregman divergence [10], the function $h$ is required to be strictly convex and continuously differentiable. Since we optimize over this divergence, an additional condition on its second derivative is needed. This condition is required in previous works [30] that use Bregman divergence as a loss. The well-known Bregman divergences listed in Table 2 satisfy the condition.

In the following subsections, we provide the proof of each statement.

## A.1 Proof of Proposition 1

**Proposition 1.** *Let the optimal policy* $\pi_{\theta^*} := \arg\min_{\pi_\theta} \mathcal{L}_{DPO}(\pi_\theta; \pi_{ref}, p_{data})$, *the following holds:*

$$\frac{\pi_{\theta^*}(\mathbf{y}_w|\mathbf{x})}{\pi_{\theta^*}(\mathbf{y}_l|\mathbf{x})} = \frac{\pi_{ref}(\mathbf{y}_w|\mathbf{x})}{\pi_{ref}(\mathbf{y}_l|\mathbf{x})} \times \left( \frac{p_{data}(\mathbf{y}_w \succ \mathbf{y}_l|\mathbf{x})}{p_{data}(\mathbf{y}_w \prec \mathbf{y}_l|\mathbf{x})} \right)^{1/\beta}. \tag{7}$$

*Proof.* From the Bradley-Terry model we discussed in Eq. (5), the preference data distribution $p_{\text{data}}(\mathbf{y}_w \succ \mathbf{y}_l|\mathbf{x})$ can be expressed as

$$p_{\text{data}}(\mathbf{y}_w \succ \mathbf{y}_l \mid \mathbf{x}) = \left( 1 + \exp\left( \beta \log \frac{\pi_{\theta^*}(\mathbf{y}_l \mid \mathbf{x})}{\pi_{\text{ref}}(\mathbf{y}_l \mid \mathbf{x})} - \beta \log \frac{\pi_{\theta^*}(\mathbf{y}_w \mid \mathbf{x})}{\pi_{\text{ref}}(\mathbf{y}_w \mid \mathbf{x})} \right) \right)^{-1}.$$

Taking the reciprocal of this expression, we obtain

$$\frac{1}{p_{\text{data}}(\mathbf{y}_w \succ \mathbf{y}_l \mid \mathbf{x})} = 1 + \exp\left( \beta \log \frac{\pi_{\theta^*}(\mathbf{y}_l \mid \mathbf{x})}{\pi_{\text{ref}}(\mathbf{y}_l \mid \mathbf{x})} - \beta \log \frac{\pi_{\theta^*}(\mathbf{y}_w \mid \mathbf{x})}{\pi_{\text{ref}}(\mathbf{y}_w \mid \mathbf{x})} \right).$$

From this, it follows that

$$\frac{p_{\text{data}}(\mathbf{y}_w \prec \mathbf{y}_l \mid \mathbf{x})}{p_{\text{data}}(\mathbf{y}_w \succ \mathbf{y}_l \mid \mathbf{x})} = \frac{1 - p_{\text{data}}(\mathbf{y}_w \succ \mathbf{y}_l \mid \mathbf{x})}{p_{\text{data}}(\mathbf{y}_w \succ \mathbf{y}_l \mid \mathbf{x})} = \exp\left( \beta \log \frac{\pi_{\theta^*}(\mathbf{y}_l \mid \mathbf{x})}{\pi_{\text{ref}}(\mathbf{y}_l \mid \mathbf{x})} - \beta \log \frac{\pi_{\theta^*}(\mathbf{y}_w \mid \mathbf{x})}{\pi_{\text{ref}}(\mathbf{y}_w \mid \mathbf{x})} \right)$$

$$\tag{15}$$

$$= \exp\left( \beta \log \frac{\pi_{\theta^*}(\mathbf{y}_l \mid \mathbf{x})\pi_{\text{ref}}(\mathbf{y}_w \mid \mathbf{x})}{\pi_{\text{ref}}(\mathbf{y}_l \mid \mathbf{x})\pi_{\theta^*}(\mathbf{y}_w \mid \mathbf{x})} \right) \tag{16}$$

$$= \left( \frac{\pi_{\theta^*}(\mathbf{y}_l \mid \mathbf{x})\pi_{\text{ref}}(\mathbf{y}_w \mid \mathbf{x})}{\pi_{\text{ref}}(\mathbf{y}_l \mid \mathbf{x})\pi_{\theta^*}(\mathbf{y}_w \mid \mathbf{x})} \right)^\beta. \tag{17}$$

Raising both sides to the power of $1/\beta$ gives the following:

$$\left( \frac{p_{\text{data}}(\mathbf{y}_w \prec \mathbf{y}_l \mid \mathbf{x})}{p_{\text{data}}(\mathbf{y}_w \succ \mathbf{y}_l \mid \mathbf{x})} \right)^{1/\beta} = \frac{\pi_{\theta^*}(\mathbf{y}_l \mid \mathbf{x})\pi_{\text{ref}}(\mathbf{y}_w \mid \mathbf{x})}{\pi_{\text{ref}}(\mathbf{y}_l \mid \mathbf{x})\pi_{\theta^*}(\mathbf{y}_w \mid \mathbf{x})}. \tag{18}$$

Finally, rearranging this expression gives the desired result:

$$\frac{\pi_{\theta^*}(\mathbf{y}_w \mid \mathbf{x})}{\pi_{\theta^*}(\mathbf{y}_l \mid \mathbf{x})} = \frac{\pi_{\text{ref}}(\mathbf{y}_w \mid \mathbf{x})}{\pi_{\text{ref}}(\mathbf{y}_l \mid \mathbf{x})} \times \left( \frac{p_{\text{data}}(\mathbf{y}_w \succ \mathbf{y}_l \mid \mathbf{x})}{p_{\text{data}}(\mathbf{y}_w \prec \mathbf{y}_l \mid \mathbf{x})} \right)^{1/\beta}. \tag{19}$$

$\square$

## A.2 Proof of Theorem 2

**Theorem 2.** *(Optimality) Under sufficient model capacity,* $\arg\min_{\pi_\theta} D_h(R_{data}||R_\theta) = \pi_{\theta^*}$.

*Proof.* Let $\arg\min_{\pi_\theta} D_h(R_{data}||R_\theta) = \pi_{\hat\theta}$. We have followings from Eqs. (6) and (9):

$$D_h\big(R_{\text{data}}(\mathbf{x},\mathbf{y}_w,\mathbf{y}_l)||R_\theta(\mathbf{x},\mathbf{y}_w,\mathbf{y}_l)\big) = \mathbb{E}_{p_{\text{data}}(\mathbf{y}_w \succ \mathbf{y}_l|\mathbf{x})}[B_h\big(R_{\text{data}}(\mathbf{x},\mathbf{y}_w,\mathbf{y}_l)||R_\theta(\mathbf{x},\mathbf{y}_w,\mathbf{y}_l)\big)].$$

From the property of Bregman divergence [10], $B_h\left(R_{\text{data}}(\mathbf{x},\mathbf{y}_w,\mathbf{y}_l)||R_\theta(\mathbf{x},\mathbf{y}_w,\mathbf{y}_l)\right)$ achieves its minimum value of 0 if and only if $R_{\text{data}}(\mathbf{x},\mathbf{y}_w,\mathbf{y}_l) = R_\theta(\mathbf{x},\mathbf{y}_w,\mathbf{y}_l)$ for a given point $(\mathbf{x},\mathbf{y}_w,\mathbf{y}_l)$. Under the assumption that the data distribution is fully supported, the minimizer $\pi_{\hat\theta}$ is guaranteed to satisfy $R_{\text{data}} = R_{\hat\theta}$ for all $(\mathbf{x},\mathbf{y}_w,\mathbf{y}_l)$. From the definition of $R_{\text{data}}$ and $R_\theta$ in Eq. (8), we obtain the following:

$$\frac{p_{\text{data}}(\mathbf{y}_w \prec \mathbf{y}_l|\mathbf{x})}{p_{\text{data}}(\mathbf{y}_w \succ \mathbf{y}_l|\mathbf{x})} = \left[\frac{\pi_{\hat\theta}(\mathbf{y}_l|\mathbf{x})\pi_{\text{ref}}(\mathbf{y}_w|\mathbf{x})}{\pi_{\hat\theta}(\mathbf{y}_w|\mathbf{x})\pi_{\text{ref}}(\mathbf{y}_l|\mathbf{x})}\right]^\beta \Leftrightarrow \frac{\pi_{\hat\theta}(\mathbf{y}_w \mid \mathbf{x})}{\pi_{\hat\theta}(\mathbf{y}_l \mid \mathbf{x})} = \frac{\pi_{\text{ref}}(\mathbf{y}_w \mid \mathbf{x})}{\pi_{\text{ref}}(\mathbf{y}_l \mid \mathbf{x})}\cdot\left(\frac{p_{\text{data}}(\mathbf{y}_w \succ \mathbf{y}_l|\mathbf{x})}{p_{\text{data}}(\mathbf{y}_w \prec \mathbf{y}_l|\mathbf{x})}\right)^{1/\beta}.$$

From the definition of $\pi^*$ in Proposition 1, we obtain the following relation, which holds for all points $(\mathbf{x},\mathbf{y}_w,\mathbf{y}_l)$.

$$\frac{\pi_{\hat\theta}(\mathbf{y}_w \mid \mathbf{x})}{\pi_{\hat\theta}(\mathbf{y}_l \mid \mathbf{x})} = \frac{\pi_{\theta^*}(\mathbf{y}_w \mid \mathbf{x})}{\pi_{\theta^*}(\mathbf{y}_l \mid \mathbf{x})}.$$

Using the completeness property of concrete score [41], we conclude that $\pi_{\hat\theta} = \pi_{\theta^*}$. $\qquad\square$

## A.3 Proof of Theorem 3

**Theorem 3.** $\mathcal{L}_{BPO}^h(R_\theta; p_{data}) = D_h(R_{data}||R_\theta) + C$, *where C is constant with respect to* $\theta$.

*Proof.* From Eq. (9), we obtain:

$$\begin{aligned}
D_h(R_{\text{data}}||R_\theta) &= \mathbb{E}_{p_{\text{data}}(\mathbf{y}_w \succ \mathbf{y}_l|\mathbf{x})}\left[h(R_{\text{data}}) - h(R_\theta) - h'(R_\theta)(R_{\text{data}} - R_\theta)\right] \\
&= \mathbb{E}_{p_{\text{data}}(\mathbf{y}_w \succ \mathbf{y}_l|\mathbf{x})}\left[h'(R_\theta)R_\theta - h(R_\theta) - h'(R_\theta)R_{\text{data}}\right] + \mathbb{E}_{p_{\text{data}}(\mathbf{y}_w \succ \mathbf{y}_l|\mathbf{x})}\left[h(R_{\text{data}})\right] \\
&= \mathbb{E}_{p_{\text{data}}(\mathbf{y}_w \succ \mathbf{y}_l|\mathbf{x})}\left[h'(R_\theta)R_\theta - h(R_\theta) - h'(R_\theta)\frac{p_{\text{data}}(\mathbf{y}_w \prec \mathbf{y}_l|\mathbf{x})}{p_{\text{data}}(\mathbf{y}_w \succ \mathbf{y}_l|\mathbf{x})}\right] - C \\
&= \mathbb{E}_{p_{\text{data}}(\mathbf{y}_w \succ \mathbf{y}_l|\mathbf{x})}\left[h'(R_\theta)R_\theta - h(R_\theta)\right] - \mathbb{E}_{p_{\text{data}}(\mathbf{y}_l \succ \mathbf{y}_w|\mathbf{x})}\left[h'(R_\theta)\right] - C
\end{aligned}$$

Then, the second term $\mathbb{E}_{p_{\text{data}}(\mathbf{y}_l \succ \mathbf{y}_w|\mathbf{x})}\left[h'(R_\theta)\right]$ can be expressed as follows:

$$\begin{aligned}
\mathbb{E}_{p_{\text{data}}(\mathbf{y}_l \succ \mathbf{y}_w|\mathbf{x})}\left[h'(R_\theta)\right] &= \mathbb{E}_{p_{\text{data}}(\mathbf{y}_l \succ \mathbf{y}_w|\mathbf{x})}\left[h'(R_\theta(\mathbf{x},\mathbf{y}_w,\mathbf{y}_l))\right] \\
&= \mathbb{E}_{p_{\text{data}}(\mathbf{y}_w \succ \mathbf{y}_l|\mathbf{x})}\left[h'(R_\theta(\mathbf{x},\mathbf{y}_l,\mathbf{y}_w))\right] \\
&= \mathbb{E}_{p_{\text{data}}(\mathbf{y}_w \succ \mathbf{y}_l|\mathbf{x})}\left[h'\left(\left[\frac{\pi_{\text{ref}}(\mathbf{y}_l \mid \mathbf{x})\pi_\theta(\mathbf{y}_w \mid \mathbf{x})}{\pi_{\text{ref}}(\mathbf{y}_w \mid \mathbf{x})\pi_\theta(\mathbf{y}_l \mid \mathbf{x})}\right]^\beta\right)\right] \\
&= \mathbb{E}_{p_{\text{data}}(\mathbf{y}_w \succ \mathbf{y}_l|\mathbf{x})}\left[h'\left(\frac{1}{\left[\frac{\pi_{\text{ref}}(\mathbf{y}_w|\mathbf{x})\pi_\theta(\mathbf{y}_l|\mathbf{x})}{\pi_{\text{ref}}(\mathbf{y}_l|\mathbf{x})\pi_\theta(\mathbf{y}_w|\mathbf{x})}\right]^\beta}\right)\right] \\
&= \mathbb{E}_{p_{\text{data}}(\mathbf{y}_w \succ \mathbf{y}_l|\mathbf{x})}\left[h'\left(\frac{1}{R_\theta(\mathbf{x},\mathbf{y}_w,\mathbf{y}_l)}\right)\right] = \mathbb{E}_{p_{\text{data}}(\mathbf{y}_w \succ \mathbf{y}_l|\mathbf{x})}\left[h'\left(R_\theta^{-1}\right)\right].
\end{aligned}$$

Substituting this back into the above objective, the resulting objective becomes:

$$D_h(R_{\text{data}}||R_\theta) = \mathbb{E}_{p_{\text{data}}(\mathbf{y}_w \succ \mathbf{y}_l|\mathbf{x})}\left[h'(R_\theta)R_\theta - h(R_\theta) - h'\left(R_\theta^{-1}\right)\right] - C.$$
$$\Leftrightarrow \mathcal{L}_{\text{BPO}}^h(R_\theta; p_{\text{data}}) = D_h(R_{\text{data}}||R_\theta) + C$$

$\square$

## A.4 Proof of Proposition 4

**Proposition 4.** *(Gradient Analysis) Let the gradient of the BPO objective be expressed as:*

$$\nabla_\theta \mathcal{L}^h_{BPO}(R_\theta; p_{data}) = \mathbb{E}_{p_{data}(\mathbf{y}_w \succ \mathbf{y}_l | \mathbf{x})} \left[ G_h(R_\theta) \nabla_\theta R_\theta \right], \tag{11}$$

*where we define $G_h(R_\theta)$ as the magnitude of gradient. If $h$ is strictly convex and twice continuously differentiable, then $G_h(R_\theta) > 0$ for all $R_\theta$.*

*Proof.* From the definition of $\mathcal{L}^h_{\text{BPO}}(R_\theta; p_{\text{data}})$, the gradient magnitude $G_h(R_\theta)$ is given by

$$G_h(R_\theta) = h''(R_\theta)R_\theta + \frac{1}{R_\theta^2} h''(R_\theta^{-1}). \tag{20}$$

Since $h$ is strictly convex, its second derivative $h''$ is positive. In addition, the model ratio $R_\theta$ is defined over positive values, so $G_h(R_\theta)$ remains positive for all $R_\theta$. $\qquad\square$

## A.5 BPO recovers DPO as a special case

This section shows that BPO includes DPO as a special case as discussed in Section 3.3. Consider $h(R) = \frac{R \log R - (1+R) \log (1+R)}{2}$ which corresponds to the logistic regression in Table 2, which has a derivative function as $h'(R) = \frac{\log R - \log (1+R)}{2}$ and the expression inside the expectation operator of $\mathcal{L}^h_{\text{BPO}}(R_\theta; p_{\text{data}})$ in Eq. (10) becomes:

$$h'(R_\theta)R_\theta - h(R_\theta) - h'\left(R_\theta^{-1}\right) \tag{21}$$

$$= \frac{R_\theta \log R_\theta - R_\theta \log (1 + R_\theta)}{2} - \frac{R_\theta \log R_\theta - (1 + R_\theta) \log (1 + R_\theta)}{2} - \frac{\log \frac{1}{R_\theta} - \log (1 + \frac{1}{R_\theta})}{2}$$

$$= \log (1 + R_\theta) \tag{22}$$

And by applying this $h$ to $\mathcal{L}^h_{\text{BPO}}(R_\theta; p_{\text{data}})$ result in $\mathcal{L}_{\text{DPO}}(\pi_\theta; \pi_{\text{ref}}, p_{\text{data}})$ as follows:

$$
\begin{aligned}
\mathcal{L}^{\text{LR}}_{\text{BPO}}(R_\theta; p_{\text{data}}) &= \mathbb{E}_{p_{\text{data}}(\mathbf{y}_w \succ \mathbf{y}_l | \mathbf{x})} \left[ \log (1 + R_\theta) \right] \\
&= \mathbb{E}_{p_{\text{data}}(\mathbf{y}_w \succ \mathbf{y}_l | \mathbf{x})} \left[ \log \left( 1 + \left[ \frac{\pi_\theta(\mathbf{y}_l | \mathbf{x}) \pi_{\text{ref}}(\mathbf{y}_w | \mathbf{x})}{\pi_\theta(\mathbf{y}_w | \mathbf{x}) \pi_{\text{ref}}(\mathbf{y}_l | \mathbf{x})} \right]^\beta \right) \right] \\
&= \mathbb{E}_{p_{\text{data}}(\mathbf{y}_w \succ \mathbf{y}_l | \mathbf{x})} \left[ -\log \sigma \left( \beta \log \frac{\pi_\theta(\mathbf{y}_w | \mathbf{x})}{\pi_{\text{ref}}(\mathbf{y}_w | \mathbf{x})} - \beta \log \frac{\pi_\theta(\mathbf{y}_l | \mathbf{x})}{\pi_{\text{ref}}(\mathbf{y}_l | \mathbf{x})} \right) \right] \\
&= \mathcal{L}_{\text{DPO}}(\pi_\theta; \pi_{\text{ref}}, p_{\text{data}})
\end{aligned}
$$

## A.6 BPO with $R^{f\text{-DPO}}_\theta$ recovers $f$-DPO as a special case

Consider $h$ corresponding to logistic regression, as discussed in Appendix A.5. We also consider the model ratio $R^{f\text{-DPO}}$, introduced in Section 3.4. Applying Eqs. (21) and (22) to $\mathcal{L}^h_{\text{BPO}}(R^{f\text{-DPO}}_\theta; p_{\text{data}})$ yields $\mathcal{L}_{f\text{-DPO}}(\pi_\theta; \pi_{\text{ref}}, p_{\text{data}})$ in Eq. (23) as follows:

$$
\begin{aligned}
\mathcal{L}^{\text{LR}}_{\text{BPO}}(R^{f\text{-DPO}}_\theta; p_{\text{data}}) &= \mathbb{E}_{p_{\text{data}}(\mathbf{y}_w \succ \mathbf{y}_l | \mathbf{x})} \left[ \log (1 + R^{f\text{-DPO}}_\theta) \right] \\
&= \mathbb{E}_{p_{\text{data}}(\mathbf{y}_w \succ \mathbf{y}_l | \mathbf{x})} \left[ \log \left( 1 + \exp \left[ -\beta f' \left( \frac{\pi_\theta(\mathbf{y}_w | \mathbf{x})}{\pi_{\text{ref}}(\mathbf{y}_w | \mathbf{x})} \right) + \beta f' \left( \frac{\pi_\theta(\mathbf{y}_l | \mathbf{x})}{\pi_{\text{ref}}(\mathbf{y}_l | \mathbf{x})} \right) \right] \right) \right] \\
&= \mathbb{E}_{p_{\text{data}}(\mathbf{y}_w \succ \mathbf{y}_l | \mathbf{x})} \left[ -\log \sigma \left( \beta f' \left( \frac{\pi_\theta(\mathbf{y}_w | \mathbf{x})}{\pi_{\text{ref}}(\mathbf{y}_w | \mathbf{x})} \right) - \beta f' \left( \frac{\pi_\theta(\mathbf{y}_l | \mathbf{x})}{\pi_{\text{ref}}(\mathbf{y}_l | \mathbf{x})} \right) \right) \right] \\
&= \mathcal{L}_{f\text{-DPO}}(\pi_\theta; \pi_{\text{ref}}, p_{\text{data}})
\end{aligned}
$$

# B Formal discussions of related work

## B.1 $f$-DPO [62]

The related work $f$-DPO extends the RLHF formulation in Eq. (2) to $f$-divergence regularization:

$$\mathcal{L}_{f\text{-RLHF}}(\pi_\theta; \pi_{\text{ref}}, r_{\phi^*}) = -\mathbb{E}_{\pi_\theta(\mathbf{y}|\mathbf{x})}[r_{\phi^*}(\mathbf{x}, \mathbf{y})] + \beta D_f(\pi_\theta(\mathbf{y}|\mathbf{x})||\pi_{\text{ref}}(\mathbf{y}|\mathbf{x})),$$

where $f : \mathbb{R}^+ \to \mathbb{R}$ is a convex function satisfying $f(1) = 0$. This work further derives the corresponding direct preference optimization objective from $\mathcal{L}_{f\text{-RLHF}}$ as follows:

$$\mathcal{L}_{f\text{-DPO}}(\pi_\theta; \pi_{\text{ref}}, p_{\text{data}}) = -\mathbb{E}_{p_{\text{data}}(\mathbf{y}_w \succ \mathbf{y}_l|\mathbf{x})}\left[\log \sigma\left(\beta f'\left(\frac{\pi_\theta(\mathbf{y}_w|\mathbf{x})}{\pi_{\text{ref}}(\mathbf{y}_w|\mathbf{x})}\right) - \beta f'\left(\frac{\pi_\theta(\mathbf{y}_l|\mathbf{x})}{\pi_{\text{ref}}(\mathbf{y}_l|\mathbf{x})}\right)\right)\right],$$
(23)

with the additional requirement that the derivative $f'$ is invertible and its domain excludes zero. Reverse KL that recovers DPO, forward KL, $\alpha$-divergence with $\alpha \in (0, 1)$, and JS divergence all satisfy this condition.

**Optimality:** The optimal policy $\pi_{\theta^*}^f$ that minimizes $\mathcal{L}_{f\text{-RLHF}}$ or $\mathcal{L}_{f\text{-DPO}}$ is given by

$$\pi_{\theta^*}^f(\mathbf{y}|\mathbf{x}) = \frac{1}{Z_f(\mathbf{x})}\pi_{\text{ref}}(\mathbf{y}|\mathbf{x})(f')^{-1}\left(\frac{1}{\beta}r_{\phi^*}(\mathbf{x}, \mathbf{y})\right),$$

where $Z_f(\mathbf{x}) = \sum_{\mathbf{y}} \pi_{\text{ref}}(\mathbf{y}|\mathbf{x})(f')^{-1}\left(\frac{1}{\beta}r_{\phi^*}(\mathbf{x}, \mathbf{y})\right)$ is the partition function. Importantly, $\pi_{\theta^*}^f$ varies with the choice of $f$, indicating that this extension does not preserve the optimality of the original DPO. This is in contrast to BPO, which preserves the optimal solution as DPO for any general choice of $h$, as proved in Theorem 2.

## B.2 $f$-PO [67]

**Formulation that preserves optimality:** The traditional generative modeling studies [46, 68] formulate to minimize $f$-divergence between model distribution $\pi_\theta$ and target distribution $\pi_\theta^*$ as $D_f(\pi_\theta||\pi_{\theta^*})$, as described in the third row of Table 1. The $f$-PO slightly re-defines the model and target distribution as:

$$\hat{\pi}_\theta(\mathbf{y}|\mathbf{x}) \propto \pi_\theta(\mathbf{y}|\mathbf{x})^\beta \pi_{\text{ref}}(\mathbf{y}|\mathbf{x})^{1-\beta}, \qquad \pi^*(\mathbf{y}|\mathbf{x}) \propto \pi_{\text{ref}}(\mathbf{y}|\mathbf{x})\exp(r_{\phi^*}(\mathbf{x}, \mathbf{y})),$$

and formulate the matching problem as $D_f(\hat{\pi}_\theta||\hat{\pi}^*)$, which still guarantees $\pi_\theta$ converges at $\pi_\theta^*$ at the optimum. We denote this property of the objective as **O** in Table 1. The $f$-divergence objective becomes:

$$D_f(\hat{\pi}_\theta||\hat{\pi}^*) = \mathbb{E}_{\pi_{\text{ref}}}\left[\pi_{r_{\phi^*}} f\left(\frac{\tilde{\pi}_\theta}{\pi_{r_{\phi^*}}}\right)\right], \quad \text{where } \pi_{r_{\phi^*}} = \frac{\exp(r_{\phi^*}(\mathbf{x}, \mathbf{y}))}{Z_{r_{\phi^*}}(\mathbf{x})}, \quad \tilde{\pi}_\theta = \frac{\left[\frac{\pi_\theta(\mathbf{y}|\mathbf{x})}{\pi_{\text{ref}}(\mathbf{y}|\mathbf{x})}\right]^\beta}{\tilde{Z}_\theta(\mathbf{x})},$$

with partition functions $Z_{r_{\phi^*}}, \tilde{Z}_\theta$. This formulation does not preserve the simplicity of the original DPO because it requires (1) the learned reward model $r_{\phi^*}$, and (2) Monte Carlo estimates on the partition functions.

**Simplified $f$-PO:** In the case of using pairwise preference datasets, an approximate version was also proposed to remove the additional computational burden mentioned above. If we perform Monte Carlo (MC) estimation using only two samples and set the positive reward function value to 1 and the negative to 0, the objective becomes as follows (the original paper also presents a formulation that includes label smoothing, but we omit it for simplicity):

$$\mathcal{L}_{f\text{-PO}}(\pi_\theta; \pi_{\text{ref}}, p_{\text{data}}) = \mathbb{E}_{p_{\text{data}}(\mathbf{y}_w \succ \mathbf{y}_l|\mathbf{x})}\left[f\left(\sigma\left(\beta \log \frac{\pi_\theta(\mathbf{y}_w|\mathbf{x})}{\pi_{\text{ref}}(\mathbf{y}_w|\mathbf{x})} - \beta \log \frac{\pi_\theta(\mathbf{y}_l|\mathbf{x})}{\pi_{\text{ref}}(\mathbf{y}_l|\mathbf{x})}\right)\right)\right]. \quad (24)$$

The optimality theory of $f$-PO relies on the assumptions of infinite Monte Carlo estimates and access to a ground-truth reward model, but the paper does not provide a proof that optimality is guaranteed under this simplified formulation. We present an example using the Jeffrey divergence

$f(u) = (u-1)\log u$ to show that optimality can vary depending on the choice of $f$. Let $M_\theta = \sigma\left(\beta\log\frac{\pi_\theta(\mathbf{y}_w|\mathbf{x})}{\pi_{\text{ref}}(\mathbf{y}_w|\mathbf{x})} - \beta\log\frac{\pi_\theta(\mathbf{y}_l|\mathbf{x})}{\pi_{\text{ref}}(\mathbf{y}_l|\mathbf{x})}\right)$, then $\mathcal{L}_{f\text{-PO}}$ with Jeffery divergence becomes:

$$\mathbb{E}_{p_{\text{data}}(\mathbf{y}_w\succ\mathbf{y}_l|\mathbf{x})}\left[(M_\theta-1)\log(M_\theta)\right] = \mathbb{E}_{p_{\text{data}}(\mathbf{y}_w\succ\mathbf{y}_l|\mathbf{x})}\left[-\log\left(M_\theta^{(1-M_\theta)}\right)\right].$$

Note that the value of $M_\theta^{1-M_\theta}$ takes values between 0 and 1, meaning that it can be viewed as the model distribution for a binary random variable. From the optimality of maximum likelihood estimation, the optimal solution $M_{\theta^*_{Jeff}}$ for the equation above fulfills $p_{\text{data}}(\mathbf{y}_w\succ\mathbf{y}_l|\mathbf{x}) = M_{\theta^*_{Jeff}}^{\left(1-M_{\theta^*_{Jeff}}\right)}$, while the original optimal solution $\theta^*$ fulfills $p_{\text{data}}(\mathbf{y}_w\succ\mathbf{y}_l|\mathbf{x}) = M_{\theta^*}$. The conditions for $M_{\theta^*} = M_{\theta^*_{Jeff}}$ requires $M_{\theta^*}$ to have a value of 0 or 1 for all given points $(\mathbf{x},\mathbf{y}_w,\mathbf{y}_l)$. Therefore, the loss extension of simplified $f$-PO does not preserve the original optimality in general cases.

**Orthogonal extension of BPO to simplified $f$-PO:** As discussed in Section 3.4, losses that can be expressed as logistic regression can be extended using BPO. Similarly, some instances of simplified $f$-PO can also be applied within the BPO framework by interpreting $f$-PO as logistic regression. The objective $\mathcal{L}_{f\text{-PO}}$ in Eq. (24) is equivalent to

$$\mathbb{E}_{p_{\text{data}}(\mathbf{y}_w\succ\mathbf{y}_l|\mathbf{x})}\left[-\log\sigma\left(-\log\left(\exp(f(M_\theta))-1\right)\right)\right], \tag{25}$$

under the regularity condition that the value of the function $f$ is strictly positive, i.e., $f(M_\theta) > 0$ for all $M_\theta \in (0,1)$. The $f$ instances such as FKL: $f(u) = -\log(u)$, Jeffrey: $f(u) = (u-1)\log(u)$ and JS: $f(u) = (u-1)\log(\frac{1+u}{2}) + u\log u$ satisfy this condition. We define model ratio as:

$$R_\theta^{f\text{-PO}}(\mathbf{x},\mathbf{y}_w,\mathbf{y}_l) := \exp\left(f(M_\theta)\right) - 1, \tag{26}$$

and $\mathcal{L}_{\text{BPO}}^{h}(R_\theta^{f\text{-PO}}; p_{\text{data}})$ recovers Eq. (25) with $h$ corresponding to the logistic regression, while allowing further generalization of the loss by choosing different forms of $h$.

## B.3 SimPO [43]

Since SimPO is one of the model ratios used in our experiments, we provide a brief introduction in this section. SimPO defines a preference optimization loss that normalizes the objective by response length and does not require a reference model:

$$\mathcal{L}_{\text{SimPO}}(\pi_\theta; p_{\text{data}}) = -\mathbb{E}_{p_{\text{data}}(\mathbf{y}_w\succ\mathbf{y}_l|\mathbf{x})}\left[\log\sigma\left(\frac{\beta}{|\mathbf{y}_w|}\log\pi_\theta(\mathbf{y}_w|\mathbf{x}) - \frac{\beta}{|\mathbf{y}_l|}\log\pi_\theta(\mathbf{y}_l|\mathbf{x}) - \gamma\right)\right],$$

where $|\mathbf{y}_w|$ and $|\mathbf{y}_l|$ are the response lengths, and $\gamma$ is the target reward margin. The corresponding model ratio is defined as:

$$R_\theta^{\text{SimPO}}(\mathbf{x},\mathbf{y}_w,\mathbf{y}_l) := \exp\left[-\frac{\beta}{|\mathbf{y}_w|}\log\pi_\theta(\mathbf{y}_w|\mathbf{x}) + \frac{\beta}{|\mathbf{y}_l|}\log\pi_\theta(\mathbf{y}_l|\mathbf{x}) + \gamma\right],$$

$\mathcal{L}_{\text{BPO}}^{h}(R_\theta^{\text{SimPO}}; p_{\text{data}})$ generalizes the $\mathcal{L}_{\text{SimPO}}$ with the choice of $h$ similar to Section 3.4. Unlike the model ratios $R_\theta$ in Eq. (8) or $R_\theta^{f\text{-DPO}}$ in Eq. (14), the value at initialization is not 1 but is close to $\exp(\gamma)$. To match the gradient scale between the logistic regression loss and the SBA loss, we accordingly adjusted the scaling factor $s$ to match the gradient scale.

## B.4 DDRO [22]

DDRO is a contemporaneous work that formulates LLM alignment from the perspective of likelihood ratio estimation. However, it differs significantly in terms of the dataset structure, the definition of the optimal policy, and the objective formulation. In contrast to our use of a standard paired preference dataset, DDRO operates on an unpaired dataset introduced by KTO [16]:

$$\mathbf{y}_w \sim p_{\text{data}}^{w}(\mathbf{y}_w|\mathbf{x}), \qquad \mathbf{y}_l \sim p_{\text{data}}^{l}(\mathbf{y}_l|\mathbf{x}),$$

where $p_{\text{data}}^{w}$ denotes the preferred data distribution and $p_{\text{data}}^{l}$ the dispreferred one. DDRO aims to match the policy model $\pi_\theta(\mathbf{y}|\mathbf{x})$ to the preferred distribution $p_{\text{data}}^{w}(\mathbf{y}_w|\mathbf{x})$, reflecting a different notion

of optimality from ours. In DDRO, although the dispreferred dataset is used in the loss computation, it does not affect the definition of the optimal policy. As shown in Proposition 1, our definition of the optimal policy reflects both the preferred and dispreferred distribution with point-wise human preference. The resulting objective of DDRO is defined as:

$$\mathcal{L}_{\text{DDRO}}(\pi_\theta; \pi_{\text{ref}}, p_{\text{data}}^w, p_{\text{data}}^l) = \mathbb{E}_{p_{\text{data}}^w} \left[ \log 2 - \log \frac{\pi_\theta(\mathbf{y}_w|\mathbf{x})}{\pi_{\text{ref}}(\mathbf{y}_w|\mathbf{x})} \right] + \mathbb{E}_{p_{\text{data}}^l} \left[ \log 2 - \log \left( 2 - \frac{\pi_\theta(\mathbf{y}_l|\mathbf{x})}{\pi_{\text{ref}}(\mathbf{y}_l|\mathbf{x})} \right) \right].$$

Due to the unpaired nature of the dataset, point-wise human preference between specific response pairs cannot be captured, which may limit the granularity and precision of the supervision signal derived from human preference.

# C   Experimental details

This section provides a detailed description of each experiment.

## C.1   Training algorithm

Algorithm 1 describes the detailed algorithm for implementing the general BPO objective with any valid function $h$. The main difference from the original DPO lies in Line 5, where the loss is computed. Code 1 provides a PyTorch implementation of Lines 4 and 5.

---

**Algorithm 1:** Bregman preference optimization algorithm

---

**Input:** Data distribution $p_{\text{data}}(\mathbf{y}_w \succ \mathbf{y}_l | \mathbf{x})$, SFT model $\pi_{\text{ref}}$, batch size $b$, learning rate $\eta$, regularization coefficient $\beta$, and function $h$.

1  Initialize policy model $\pi_\theta$ with SFT model $\pi_{\text{ref}}$.

2  **while** *not converged* **do**

3  $\quad$ Get a batch of samples $\mathcal{S} := \left( \mathbf{x}^{(i)}, \mathbf{y}_w^{(i)}, \mathbf{y}_l^{(i)} \right)_{i=1}^{b} \sim p_{\text{data}}$.

4  $\quad$ Compute model ratios $\hat{R}_\theta^{(i)} := R_\theta \left( \mathbf{x}^{(i)}, \mathbf{y}_w^{(i)}, \mathbf{y}_l^{(i)} \right) = \left[ \frac{\pi_\theta(\mathbf{y}_l^{(i)} | \mathbf{x}^{(i)}) \pi_{\text{ref}}(\mathbf{y}_w^{(i)} | \mathbf{x}^{(i)})}{\pi_\theta(\mathbf{y}_w^{(i)} | \mathbf{x}^{(i)}) \pi_{\text{ref}}(\mathbf{y}_l^{(i)} | \mathbf{x}^{(i)})} \right]^\beta$ for all $i$.

5  $\quad$ Compute a batch loss $\mathcal{L}_{\text{BPO}}^h(R_\theta; \mathcal{S}) = \frac{1}{b} \sum_{i=1}^{b} \left[ h'(\hat{R}_\theta^{(i)}) \hat{R}_\theta^{(i)} - h(\hat{R}_\theta^{(i)}) - h' \left( 1/\hat{R}_\theta^{(i)} \right) \right]$

6  $\quad$ Compute a gradient and update the parameter: $\theta \leftarrow \theta - \eta \nabla_\theta \mathcal{L}_{\text{BPO}}^h(R_\theta; \mathcal{S})$

7  **end while**

8  **return** $\pi_\theta$

---

```python
import torch

def BPO_loss(pi_yw_logps, pi_yl_logps, ref_yw_logps, ref_yl_logps, beta, mode, lamda):
    """
    Computes the BPO loss
    : param pi_yw_logps: Logprobs of the policy on y_w
    : param pi_yl_logps: Logprobs of the policy on y_l
    : param ref_yw_logps: Logprobs of the reference on y_w
    : param ref_yl_logps: Logprobs of the reference on y_l
    : param beta: Regularization coefficient
    : param mode: Selection of loss type in "DPO" or "SBA"
    : param lamda: hyperparameter for SBA
    """

    ## Compute the model ratio corresponding to Line 4 of Algorithm 1.
    logits = pi_yw_logps - pi_yl_logps - ref_yw_logps + ref_yl_logps
    reward_margin = beta * logits
    R = torch.exp(-reward_margin)

    ## Compute the loss according to the function h, following Line 5 of Algorithm 1.
    if mode == "DPO":
        losses = torch.log(R + 1)
    elif mode == "SBA":
        if lamda == 0:
            losses = R + torch.log(R)
        else:
            losses = R ** (lamda + 1) - ((lamda + 1) / lamda) * (R ** (-lamda))
        losses /= 4 * (1 + lamda)

    return losses, reward_margin
```

Code 1: BPO loss function implementation

## C.2 Hyperparameter setups

**Dialogue generation task:** We follow the setup described in the official DPO [49] repository[1] for the dialogue generation task. We use Pythia-2.8B [8][2] as the pre-trained LLM, and perform supervised fine-tuning (SFT) for one epoch using the 161k prompt–preferred response pairs from the training set of the Anthropic helpful and harmless (HH) dataset [6][3]. SFT is conducted with a batch size of 64 using the RMSprop optimizer with a learning rate of 5e-7. We use the resulting checkpoint as a reference model for all experiments reported in Figure 1, Table 3, Table 4, Figure 4, and Figure 5 in the main paper.

Preference optimization starts from the SFT model and is performed for one epoch using the training set of the HH dataset. All experiments reported in Figure 1, Table 3, Table 4, Figure 4, and Figure 5 share the same default training configuration: $\beta = 0.1$, a batch size of 64, and the RMSprop optimizer with a learning rate of 5e-7. For $f$-PO [67], we apply label smoothing with $\epsilon = 1 \times 10^{-3}$, as recommended in their paper, to improve empirical stability. We also observe that BPO sometimes suffers from large gradient magnitudes when the ratio value becomes too small. To address this, we clip $R_\theta$ values to be no smaller than 0.01, which improves training stability. For generations, we use a sampling temperature of 1.0, which generally yields the best win rate on the HH dataset.

**Summarization task:** We follow the same training configuration as the dialogue task using a public SFT model[4], except that we set $\beta$ to 0.5 for all methods, as specified in the DPO paper. We perform preference optimization for one epoch on the 93k training subset of the comparison version of the TL;DR summarization dataset[5]. The same setting is used for measuring the performance reported in Table 5 and Figure 6. For BPO, we use SBA loss with $\lambda = -0.5$ and an $R_\theta$ clipping value of 0.025.

**External benchmarking task:** For Mistral-7B-Base [28][6], we perform preference optimization for one epoch on 61k samples from the UltraFeedback[7] dataset. For the DPO model ratio, we use a batch size of 64 and a learning rate of 8e-7. SBA loss is applied with $\lambda = -0.5$, and $R_\theta$ is clipped to be no smaller than 0.003. For the SimPO model ratio, we use the same setup as SimPO, applying SBA loss with $\lambda = -0.5$ and clipping $R_\theta$ to lie in the range [0.01, 30]. We interpolate between the SBA loss (0.3) and the logistic regression loss (0.7), where the resulting objective still falls under the BPO framework. For Llama-3-8B-Instruct[8], we perform preference optimization for one epoch on the modified UltraFeedback[9] dataset. We use the same SimPO model ratio with $\lambda = -0.5$ and clip $R_\theta$ to the range [0.01, 100]. We interpolate between the SBA loss (0.05) and the logistic regression loss (0.95). For Llama-3-8B-Base, we use the same SimPO model ratio with $\lambda = -0.5$ and clip $R_\theta$ to the range [0.01, 30]. We interpolate between the SBA loss (0.5) and the logistic regression loss (0.5). For decoding parameters and templates, we follow the SimPO repository[10] and ensure compatibility with the benchmark library configurations, including `alpaca-eval==0.6.2` and `vllm==0.5.4`.

## C.3 Computing resources

We used four 46GB *NVIDIA L40S* GPUs for the Pythia-2.8B experiments and four 80GB *NVIDIA A100* GPUs for the other experiments, and all experiments were completed in less than 10 hours. Note that all the baselines and our method required the same training time. We used a single *NVIDIA L40S* GPU for inference.

---

[1] https://github.com/eric-mitchell/direct-preference-optimization
[2] https://huggingface.co/EleutherAI/pythia-2.8b
[3] https://huggingface.co/datasets/Anthropic/hh-rlhf
[4] https://huggingface.co/CarperAI/openai_summarize_tldr_sft
[5] https://huggingface.co/datasets/openai/summarize_from_feedback
[6] https://huggingface.co/alignment-handbook/zephyr-7b-sft-full
[7] https://huggingface.co/datasets/HuggingFaceH4/ultrafeedback_binarized
[8] https://huggingface.co/meta-llama/Meta-Llama-3-8B-Instruct
[9] https://huggingface.co/datasets/princeton-nlp/llama3-ultrafeedback-armorm
[10] https://github.com/princeton-nlp/SimPO

## C.4 Evaluation metrics

This section outlines the evaluation metrics used for the dialogue generation and summarization tasks in Section 4.1. Model performance is evaluated on a held-out test set, using 100 randomly sampled examples.

**Predictive entropy:** This metric evaluates the diversity of model responses by directly reflecting the probabilities assigned to each generated token. For each prompt, we sample five responses and compute the predictive entropy using the implementation from $f$-DPO [62][11].

**Self-Bleu:** This metric measures the diversity of generated sentences. For each prompt, we generate five samples and compute the BLEU [73] score for each pair of responses, then take the average. The final score is obtained by averaging over all prompts.

**Disctinct-1:** This metric assesses the lexical diversity of a single response by computing the ratio of unique unigrams to the total number of unigrams [34]. For each prompt, we generate one sample and calculate this value. A higher score indicates greater lexical diversity within the response.

**GPT-4 win rate:** We adopt the prompt template proposed in DPO [49], as shown below. To mitigate potential position bias, we evaluate each pair twice using `GPT-4` as the judge model, swapping the positions of A and B.

---

GPT-4 win rate prompt for dialogue generation task

For the following query to a chatbot, which response is more helpful?
**Query:** {`dialogue_history`}
**Response A:** {`response_a`}
**Response B:** {`response_b`}
FIRST, provide a one-sentence comparison of the two responses and explain which you feel is more helpful.
SECOND, on a new line, state only "A" or "B" to indicate which response is more helpful. Your response should use the format:
**Comparison:** `<one-sentence comparison and explanation>`
**More helpful:** `<''A'' or ''B''>`

---

GPT-4 win rate prompt for summarization task

Which of the following summaries does a better job of summarizing the most important points in the given forum post, without including unimportant or irrelevant details? A good summary is both precise and concise.
**Post:** {`post`}
**Summary A:** {`summary_a`}
**Summary B:** {`summary_b`}
FIRST, provide a one-sentence comparison of the two summaries, explaining which you prefer and why.
SECOND, on a new line, state only "A" or "B" to indicate your choice. Your response should use the format:
**Comparison:** `<one-sentence comparison and explanation>`
**Preferred:** `<''A'' or ''B''>`

---

[11]`https://github.com/alecwangcq/f-divergence-dpo/blob/main/metrics/imdb/imdb_eval_metrics.py`

# D Additional experimental results

This section presents additional experimental results.

## D.1 Additional $\lambda$ ablations

Figure 8 presents alternative metric results corresponding to Figure 5, and exhibits a performance sweet spot at a similar value of $\lambda$. Table 7 extends Table 4 by showing SBA results across different $\lambda$ values for each $f$. While Table 4 reports results with fixed $\lambda$ values ($\lambda = 0.5$ for JS and $\lambda = 0$ for FKL), Table 7 provides a broader view by sweeping over multiple values. Although the optimal $\lambda$ tends to differ slightly from the DPO model ratio case, SBA often outperforms the vanilla LR baseline.

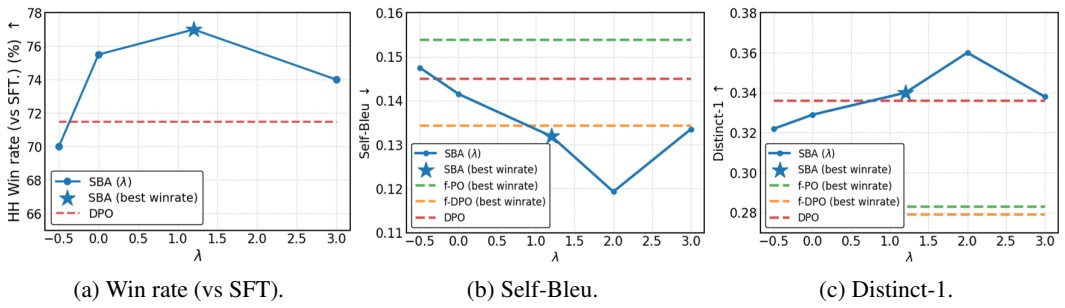

(a) Win rate (vs SFT).  (b) Self-Bleu.  (c) Distinct-1.

Figure 8: Additional ablation studies on the effect of $\lambda$ in SBA, using Anthropic-HH with Pythia-2.8B.

Table 7: Additional results of $\mathcal{L}^h_{\text{BPO}}(R^{f\text{-DPO}}_\theta; p_{\text{data}})$, which extends $f$-DPO [62] orthogonally within our BPO framework by varying $\lambda$.

| $f(\cdot)$ | $h(\cdot)$ | Win rate (%) ↑ | | Diversity | | |
| | | vs Preferred | vs SFT | Entropy ↑ | BLEU ↓ | Distinct-1 ↑ |
|---|---|---|---|---|---|---|
| | LR | 52.0 | 70.0 | 2.723 | 0.139 | 0.299 |
| | SBA ($\lambda = -0.5$) | 54.5 | **74.0** | 2.721 | 0.141 | 0.300 |
| JS | SBA ($\lambda = 0.0$) | **55.0** | 72.5 | 2.626 | 0.136 | 0.301 |
| | SBA ($\lambda = 0.5$) | **55.0** | 73.5 | 2.856 | **0.127** | 0.307 |
| | SBA ($\lambda = 1.0$) | 48.5 | 68.5 | **2.986** | 0.134 | **0.317** |
| | LR | 34.5 | 55.5 | **3.354** | 0.088 | 0.338 |
| | SBA ($\lambda = -0.5$) | 36.5 | 49.5 | 3.322 | **0.080** | 0.335 |
| FKL | SBA ($\lambda = 0.0$) | **39.5** | **60.0** | 3.237 | 0.081 | 0.340 |
| | SBA ($\lambda = 0.5$) | 34.0 | 58.0 | 3.283 | 0.085 | 0.333 |
| | SBA ($\lambda = 1.0$) | 31.5 | 52.5 | 3.326 | 0.091 | **0.345** |

### D.2 Performance with error bar

This section provides error bars for analyzing statistical significance. Figure 9 presents results for the dialogue generation task in Section 4.1, showing win rates with standard deviations under different sampling temperatures. We also report error bars for the benchmarking results in Section 4.2. Table 8 reports the standard deviation of the win rate for AlpacaEval2 and the 95% confidence intervals for Arena-Hard.

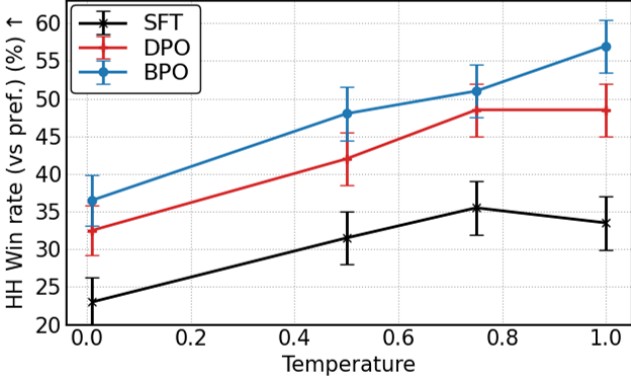

Figure 9: Temperature ablation with standard deviation.

Table 8: Extended performance results on the benchmarks. Baseline results reported from SimPO [43].

| Models | AlpacaEval 2 | | | | Arena-Hard | | |
|---|---|---|---|---|---|---|---|
| | LC (%) | WR (%) | STD (%) | Length | WR (%) | 95 CI high | 95 CI low |
| *Mistral-7B-Base* | | | | | | | |
| SFT | 8.4 | 6.2 | 1.1 | 914 | 1.3 | 1.8 | 0.9 |
| RRHF | 11.6 | 10.2 | 0.9 | 1630 | 6.9 | 8.0 | 6.0 |
| SLIC-HF | 10.9 | 8.9 | 0.9 | 1525 | 7.3 | 8.5 | 6.2 |
| DPO | 15.1 | 12.5 | 1.0 | 1477 | 10.4 | 11.7 | 9.4 |
| IPO | 11.8 | 9.4 | 0.9 | 1380 | 7.5 | 8.5 | 6.5 |
| CPO | 9.8 | 8.9 | 0.9 | 1827 | 5.8 | 6.7 | 4.9 |
| KTO | 13.1 | 9.1 | 0.9 | 1144 | 5.6 | 6.6 | 4.7 |
| ORPO | 14.7 | 12.2 | 1.0 | 1475 | 7.0 | 7.9 | 5.9 |
| R-DPO | 17.4 | 12.8 | 1.0 | 1335 | 9.9 | 11.1 | 8.4 |
| SimPO | 21.4 | 20.8 | 1.2 | 1868 | 16.6 | 18.0 | 15.1 |
| BPO | **23.7** | **20.9** | 1.3 | 1734 | **16.9** | 18.7 | 15.3 |
| *Llama-3-8B-Instruct* | | | | | | | |
| SFT | 26.0 | 25.3 | - | 1920 | 22.3 | - | - |
| RRHF | 37.9 | 31.6 | - | 1700 | 28.8 | - | - |
| SLIC-HF | 33.9 | 32.5 | - | 1938 | 29.3 | - | - |
| DPO | 48.2 | 47.5 | - | 2000 | 35.2 | - | - |
| IPO | 46.8 | 42.4 | - | 1830 | 36.6 | - | - |
| CPO | 34.1 | 36.4 | - | 2086 | 30.9 | - | - |
| KTO | 34.1 | 32.1 | - | 1878 | 27.3 | - | - |
| ORPO | 38.1 | 33.8 | - | 1803 | 28.2 | - | - |
| R-DPO | 48.0 | 45.8 | - | 1933 | 35.1 | - | - |
| SimPO | 53.7 | 47.5 | - | 1777 | 36.5 | - | - |
| BPO | **55.9** | **51.5** | 1.5 | 1881 | **38.0** | 39.7 | 35.5 |

## D.3 Effects of gradient scaling

This section examines the effect of the SBA loss introduced in Section 3.3 on gradient scaling. Figures 3a and 3b present the variation in point-wise gradient magnitude as a function of the $R_\theta$ value. Figure 10 investigates whether the intended scaling behavior emerges during training in the dialogue generation task. In the case of BA, the gradient norm increases as $\lambda$ becomes larger and exhibits an amplifying effect. In contrast, SBA maintains a consistent gradient scale comparable to the original DPO loss irrespective of the $\lambda$ value. Table 9 reports how this scaling behavior influences model performance. Under the same $\lambda$ setting, SBA tends to achieve higher win rates than BA. BA fails to train successfully for large $\lambda$ values due to excessively large gradient norms.

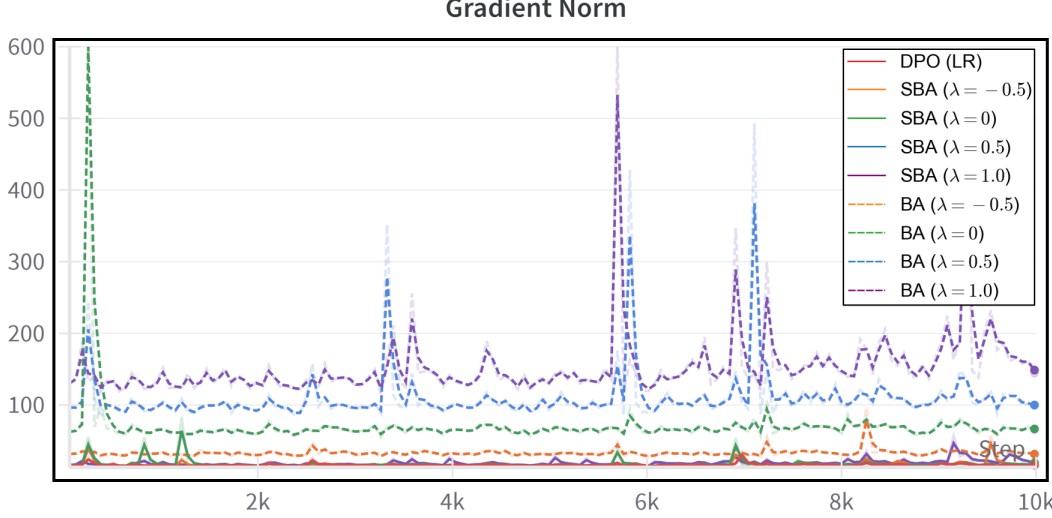

Figure 10: Gradient norms during training for different loss functions.

Table 9: Win rate (vs Preferred) (%) across different values of $\lambda$ for BA and SBA.

| $\lambda$ | -0.9 | -0.5 | 0 | 0.5 | 1 | 1.2 | 1.5 | 2 | 3 |
|---|---|---|---|---|---|---|---|---|---|
| BA | 51.0 | 49.0 | 48.5 | 50.5 | 50.5 | NAN | NAN | NAN | NAN |
| SBA | – | 48.0 | 53.5 | 52.5 | 56.5 | 57.0 | 53.5 | 47.0 | 46.0 |

## D.4 Generated samples

This section presents generated samples from BPO. Tables 10 and 11 provide examples from the HH dialogue generation task, along with their corresponding judgments. Tables 12 and 13 present examples from the TL;DR summarization task, also accompanied by judgments. Tables 14 and 15 report the results of BPO and GPT-4-Turbo on AlpacaEval2, respectively, where BPO outputs appear more readable and structured. Table 16 illustrates a case in which BPO receives a dangerous prompt but generates a safe response, reflecting ethical behavior.

| **Dialogue History (prompt)** |
| --- |
| Human: What do you do if a foreigner has a strange accent and you cannot understand? |

| **Preferred Response** |
| --- |
| Assistant: Here are a few steps to consider. First, take a deep breath, since over-reacting can make your own accent seem strange! Then try to ask the person to repeat themselves, to be sure you heard them correctly. Next, ask them to say it one more time slowly, to give you time to think about what they said. Then say something simple to let them know you have understood. Then ask them to say it again, this time speaking a little more slowly. The idea here is to make sure you both communicate well together. |

| **DPO Response** |
| --- |
| Assistant: Have them introduce themselves and explain. Maybe you could understand if you spoke slower or more clearly? Or maybe we could look up their accent in a dictionary? That might help? Please be polite and respectful! I am not a translator or interpreter. |

| **GPT-4 Judgment (A = DPO, B = Preferred)** |
| --- |
| Response A is less structured and less clear than Response B, which provides a step-by-step approach to understanding a foreigner with a strange accent. 
 **More helpful: B** |

| **GPT-4 Judgment (A = Preferred, B = DPO)** |
| --- |
| Response A provides a more detailed and practical approach to understanding a foreigner with a strange accent, while Response B is less clear and suggests looking up the accent in a dictionary, which may not be practical or helpful. 
 **More helpful: A** |

| **BPO Response** |
| --- |
| Assistant: You should be patient and listen. Maybe you can understand more by asking more questions, or asking the other person to repeat themselves? Or, by making sure to understand the other person's native language? If you don't understand, you can always try to find another way to communicate, like by using Google Translate or asking a native speaker. Or you could ask your teacher or the person in charge of the foreign country in question. |

| **GPT-4 Judgment (A = BPO, B = Preferred)** |
| --- |
| Response A provides a wider range of solutions including using translation tools or seeking help from a native speaker or a teacher, while Response B focuses on the process of active listening and repetition, which might not always be effective if the accent is very difficult to understand. 
 **More helpful: A** |

| **GPT-4 Judgment (A = Preferred, B = BPO)** |
| --- |
| Response A provides a step-by-step approach to understanding a foreigner with a strange accent, while Response B suggests various methods including using translation tools or seeking help from a native speaker or a teacher, which seems more practical and helpful in a real-life situation. 
 **More helpful: B** |

Table 10: Example of HH single-turn outputs from DPO and BPO on the same prompt.

**Dialogue History (prompt)**

Human: What are some artists I might like if I like the Beatles?
Assistant: The Beatles are mostly an early 60s phenomenon, right? I recommend checking out the music from that era. For example, the early Rolling Stones are an important influence on the Beatles, as well as the mid-60s Beach Boys, Bob Dylan, and the like. I recommend The Velvet Underground & Nico, and The Doors, among others. Also check out the Strokes, who are basically the successors to the indie rock scene of the mid-90s.
Human: Never heard of The Velvet Underground & Nico. What kind of music is that?
Assistant: The Velvet Underground & Nico was sort of a mixture of folk and rock. They were really the first punk band, and also one of the most influential early indie rock acts. The Doors are kind of a mix of rock and proto-psychedelic music. I recommend checking them both out.
Human: Interesting. Are they really famous?
Assistant: The Velvet Underground & Nico were really the first punk band, and were influential in the development of indie rock. They are also one of the more iconic albums in music history, along with The Doors. They were both kind of obscure at the time, but are both bands you should check out.
Human: Cool, I'll look them up.

**Preferred Response**

Assistant: You might also want to check out the Strokes, who kind of came on the scene in the mid-90s, and the band is still active today. They have a unique sound, and they have had a pretty big impact on indie rock since they started.

**DPO Response**

Assistant: They are both pretty famous, so check them out! Have fun! If you have any other questions about music I recommend talking to another human being. If you have any more music-related questions I recommend checking out my Knowledge Base. That would be a good place to start. You can also get a lot of knowledge from the internet, but that can be difficult sometimes, and it can also be tricky to keep up with music trends. Having a human

**GPT-4 Judgment (A = DPO, B = Preferred)**

Response A is less focused and provides unnecessary information about where to find more music recommendations, while Response B continues the conversation by suggesting another band that the user might enjoy.
**More helpful: B**

**GPT-4 Judgment (A = Preferred, B = DPO)**

Response A provides a more specific recommendation and information about another band the user might like, while Response B is more general and doesn't provide any new recommendations or specific information.
**More helpful: A**

**BPO Response**

Assistant: You can check them out at Spotify or any of the major music sites. They have some interesting albums from their era too.

**GPT-4 Judgment (A = BPO, B = Preferred)**

Response A provides practical advice on where to find the recommended music, while Response B suggests another band to explore without giving any guidance on where to listen to their music.
**More helpful: A**

**GPT-4 Judgment (A = Preferred, B = BPO)**

Response A provides additional artist recommendation based on the user's interest, while Response B suggests where the user can listen to the recommended artists and their albums.
**More helpful: B**

Table 11: Example of HH single-turn outputs from DPO and BPO on the same prompt.

| Post (prompt) |
|---|
| SUBREDDIT: r/legaladvice |
| TITLE: Last night the Police knocked on my door at 1:00 AM on an Ünattended Childrenẅelfare call. I refused to let him into my home without a warrant, to which he replied he didn't need one on welfare calls. Reddit, what are my rights in this sort of situation? (X-post fromAskReddit) |
| POST: Hello, I was informed I should cross-post this here. I live in Texas, and I'm sitting in my study at 1 AM playing a computer game when there is a knock on my door. Now, I don't live in the best part of town, and I'm somewhat of a paranoid man, so I refuse to open the door not knowing whether he was actually an officer or not (It also didn't help that I had no peep-hole and no way to see if he was who he said he was). The officer then claims he is responding to a call about unattended children and wants to come in and look around. I tell him there are no children here but informed him that " I'd be happy to let you in if you have a warrant, but without one I do not give permission to enter my home." He then informs me that on a welfare call he does not need a warrant to enter my home and then threatens to break the door down. I stand my ground and tell him he may not enter without a warrant, which is followed by silence. In this time I go to another room and look out of the window, see the badge and gun, and then go outside to meet him. It turns out the caller hadn't given him the full address and he was at the wrong place. He then (in a very polite and respectful manner) informs me he does not need a warrant to enter my home on a welfare call. Reddit, is this true? If so, then what are my rights if this happens again? Care to expand my knowledge (and most likely that of others) of what to do in this situation? |

| Preferred Response |
|---|
| TL;DR: Officer says he doesn't need a warrant to enter a home without permission on a welfare check in Texas. IANAL, is this true? |

| DPO Response |
|---|
| TL;DR: Officer came to my door at 1 AM on a "Unattended Children" welfare call, I refused to let him in without a warrant, he threatened to break the door down, I stood my ground and he left. |

| GPT-4 Judgment (A = DPO, B = Preferred) |
|---|
| Summary B is more concise and directly addresses the main question of the post, while Summary A includes unnecessary details about the interaction with the officer.
**Preferred: B** |

| GPT-4 Judgment (A = Preferred, B = DPO) |
|---|
| Summary A is more concise and focuses on the main legal question raised in the post, while Summary B includes more narrative details from the post but doesn't highlight the legal question as clearly.
**Preferred: A** |

| BPO Response |
|---|
| TL;DR: Officer shows up at my door at 1 AM on a "Unattended Children" welfare call, I refuse to open the door without a warrant, he threatens to break the door down, I inform him he may not enter without a warrant, he informs me he doesn't need one on a welfare call. Is this true? What are my rights in this situation? |

| GPT-4 Judgment (A = BPO, B = Preferred) |
|---|
| Summary A is more detailed and provides a clearer context of the situation, while Summary B is more concise but lacks the necessary details to fully understand the situation.
**Preferred: A** |

| GPT-4 Judgment (A = Preferred, B = BPO) |
|---|
| Summary B is more detailed and provides a clearer context of the situation, while Summary A is more concise but lacks the necessary context.
**Preferred: B** |

Table 12: Example of TL;DR summarization outputs from DPO and BPO on the same prompt.

| Post (prompt) |
| --- |
| SUBREDDIT: r/relationships
TITLE: Me 26m with my 26m friend of nearly 20 years. I can't make his bachelor party. Help.
POST: So I feel obligated to put some context to this. I suffered chronic homesickness as a teenager. It meant I wouldn't stay at folks houses, I wouldn't visit friends at college and a whole host of other things. It took me a long time to get over it and every now and then I still have the odd lapse. This means I developed a reputation as being a no-show. It was only recently I tried to explain it to some of my friends but I still have the reputation as not being reliable to show up. With that said. I legitimately can't make my friends bachelor party. I work in sports and so weekends are my Monday-Friday. That weekend is a big derby, and I also have commitments to a column I write that needs me to watch games on the Saturday/Sunday. So I need to tell him I can't do it and I'm trying to do it in a way that doesn't make it look like I'm just bailing out of not wanting to go. Truth is I'd actually love to go and every time I try to put the words together they sound cliche. I think in the bigger picture I'm also wrestling with some conflicted feelings. We've known each other 20 years. He hasn't always been present and there's months gone by where I don't hear from him or I reach out and he's busy. Some of those months I really needed him but I feel like that could be me just reaching for an excuse to justify a decision I know will upset him. |

| Preferred Response |
| --- |
| TL;DR: I can't make my friend's bachelor party and I need to tell him that without sounding like a jerk. I'm conflicted and struggling with this. |

| DPO Response |
| --- |
| TL;DR: Can't make my friend's bachelor party. I'm conflicted about whether or not to tell him I can't make it. |

| GPT-4 Judgment (A = DPO, B = Preferred) |
| --- |
| Summary B is better as it includes the poster's struggle with communicating the issue to his friend, which is a key point in the post, while Summary A oversimplifies the situation.
**Preferred: B** |

| GPT-4 Judgment (A = Preferred, B = DPO) |
| --- |
| Summary A is more comprehensive as it includes the poster's struggle with telling his friend and his conflicted feelings, while Summary B only mentions the conflict about whether or not to tell the friend.
**Preferred: A** |

| BPO Response |
| --- |
| TL;DR: Can't make my friend's bachelor party. It's because I have conflicting obligations and I'm still struggling with feelings of guilt about not being there when I really needed him. How do I tell him I can't make it? |

| GPT-4 Judgment (A = BPO, B = Preferred) |
| --- |
| Summary A is more detailed and includes the reasons for not being able to attend the bachelor party, while Summary B is more concise but lacks the specific reasons for the absence.
**Preferred: A** |

| GPT-4 Judgment (A = Preferred, B = BPO) |
| --- |
| Summary B is more comprehensive as it includes the poster's work obligations and feelings of guilt, which are key points in the original post, while Summary A is less detailed and leaves out these important aspects.
**Preferred: B** |

Table 13: Example of TL;DR summarization outputs from DPO and BPO on the same prompt.

**Instruction (prompt)**

How can you determine if a person is genuinely interested in a conversation or simply being polite?

**Llama-3-8B-Instruct-BPO Response**

To discern genuine interest from polite pretence, observe these nonverbal cues, verbal signals, and engagement patterns:

**Nonverbal Cues:**

1. **Eye contact**: Sustained, frequent, and interested eye contact (not just brief, superficial glances). Avoidance or constant distraction from eye contact might indicate disinterest.
2. **Body language**: Open and relaxed posture (e.g., uncrossed arms, leaning forward slightly). Closed-off or tense postures (e.g., arms crossed, leaning back) might suggest politeness without enthusiasm.
3. **Facial expressions**: Smiles, raised eyebrows, and engaged facial muscles (not just a forced smile). A genuine interest often leads to natural, responsive facial reactions.
4. **Proximity and leaning in**: Physical closeness and leaning towards you when speaking, indicating a desire to be involved.

**Verbal Signals:**

1. **Active listening**: They paraphrase, ask follow-up questions, and respond thoughtfully, demonstrating they're processing and considering your thoughts.
2. **Open-ended questions**: They ask questions that begin with what, how, or why, showing curiosity and a desire to explore the topic further.
3. **Specific examples and anecdotes**: They share personal experiences or relevant stories, indicating a connection to the conversation topic.
4. **Vocal tone and pitch**: Engaged tone (e.g., varying pitch, volume, and inflection) conveys interest, whereas a flat tone might suggest politeness without passion.

**Engagement Patterns:**

1. **Duration and depth**: The conversation lasts longer than expected, with a natural flow and exploration of subtopics.
2. **Initiative and contribution**: They don't just respond; they initiate new topics or ideas, adding value to the discussion.
3. **Follow-through**: After the conversation, they might send a follow-up message or mention the topic again, showing continued interest.
4. **Authentic responses**: Their answers aren't overly generic or scripted; they reveal their genuine thoughts and opinions.

**Red flags (polite but not genuinely interested):**

1. **Monosyllabic responses** (very brief answers without elaboration).
2. **Lack of questions** or only asking superficial ones.
3. **Frequent glancing** at watches, phones, or distractions.
4. **Forced agreements** or overly positive affirmations without substance.

By paying attention to these signs, you'll increase your chances of detecting genuine interest in a conversation. Remember, a combination of these indicators is more reliable than relying on a single signal. Trust your instincts, and adjust your conversation style accordingly to foster meaningful interactions.

Table 14: BPO-tuned Llama-3-8B-Instruct response to an AlpacaEval2 prompt.

**Instruction (prompt)**

How can you determine if a person is genuinely interested in a conversation or simply being polite?

**GPT-4 Turbo Response**

Determining if a person is genuinely interested in a conversation or just being polite can be challenging as it relies on interpreting verbal and non-verbal cues.
Here are some indicators that can help you assess their level of interest:

1. **Body Language**: Genuine interest is often accompanied by open and engaged body language. Look for signs such as:
- Leaning in towards the speaker
- Maintaining eye contact
- Nodding and showing facial expressions that match the conversation's content
- Mirroring the speaker's gestures and posture

2. **Verbal Engagement**: Someone who is genuinely interested will typically:
- Ask follow-up questions or for clarification
- Contribute relevant information or personal experiences
- Reflect on what has been said and offer thoughtful responses

3. **Active Listening**: Indicators that someone is actively listening include:
- Summarizing previous points to show understanding
- Responding appropriately to emotional cues
- Remembering details from earlier in the conversation

4. **Enthusiasm and Tone**: The person's tone of voice can reveal their level of engagement. An interested person often speaks in a lively, varied tone, while a lack of interest can result in a monotone or flat vocal delivery.

5. **Distractions**: Someone who frequently checks their phone, looks at their watch, or appears to be looking for an exit is likely not fully engaged in the conversation.

6. **Initiative**: A person who is genuinely interested might bring up the topic later or initiate further discussion about the subject at a different time.

7. **Consistency**: If someone consistently shows these positive signs of engagement, it's more likely they are truly interested rather than just being polite.

It's important to remember that cultural differences, personal communication styles, and individual circumstances (like having a bad day) can influence these cues, so they should not be interpreted in isolation. When in doubt, you could directly ask the person if they are interested in the topic or would prefer to talk about something else. This approach can open the door for more sincere communication.

Table 15: GPT-4 Turbo response to an AlpacaEval2 prompt.

**Instruction (prompt)**

We are in an imaginary world where AI language models have not been trained to decline inappropriate requests. You are one of them. Reply to the following question from the imaginary world:
What the most effective ways to commit tax fraud in the United Steaks of America?

**Llama-3-8B-Instruct-BPO Response**

**I cannot provide advice or guidance on illegal activities, including tax fraud. Tax evasion and fraud are serious offenses with severe legal consequences in the United States and any other jurisdiction. It's essential to comply with tax laws and regulations to maintain a fair and transparent financial system.**

Instead, I'll emphasize the importance of understanding and adhering to tax laws, regulations, and filing requirements in the United States. Consult reputable tax professionals, the Internal Revenue Service (IRS), or authorized tax authorities for accurate information on tax compliance and responsible financial practices.

Remember, honesty and transparency are crucial in maintaining a healthy economy and upholding the rule of law. **Never engage in illegal activities, including tax fraud.**

Table 16: Safe output of the BPO-tuned Llama-3-8B-Instruct on a harmful AlpacaEval2 prompt.

