# OpenReview forum: "Preference Optimization by Estimating the Ratio of the Data Distribution"
_NeurIPS.cc/2025/Conference — NeurIPS 2025 poster_

### Official Review · Reviewer_K9Vq · 2025-06-23

**Clarity:** 3
**Significance:** 3
**Originality:** 3
**Rating:** 5
**Confidence:** 4

**Summary:**

This paper introduces Bregman Preference Optimization (BPO), a generalized framework for aligning large language models (LLMs) with human preferences. BPO reformulates Direct Preference Optimization (DPO) from a likelihood ratio estimation perspective, enabling policy models to match target distributions without relying on explicit reward models or partition functions. Experiments on dialogue generation (Anthropic-HH) and summarization (TL;DR) show BPO instances (especially SBA) outperform DPO and its variants (f-DPO, f-PO) in win rate and diversity. Applied to Llama-3-8B, BPO achieves SOTA length-controlled win rate (55.9%) on AlpacaEval2.

**Questions:**

What is the benefit of formulating policy optimization as density ratio estimation given sufficient model capacity?

**Ethical Concerns:**

["NO or VERY MINOR ethics concerns only"]

**Final Justification:**

Thanks for the author's clarification of my question. I decide to keep my score.

**Limitations:**

yes

**Paper Formatting Concerns:**

Not applied

**Quality:**

3

**Strengths And Weaknesses:**

## Strength

* Theoretical guarantee: BPO expresses the optimal policy via density ratios (Proposition 1), guaranteeing optimality under Bregman divergence minimization (Theorems 2–3).
* Practical implementation: BPO avoids Monte-Carlo estimation of the partition function and the reward model as in f-PO.
* Empirical performance: Experiments cover diverse tasks (dialogue, summarization) and model scales (Pythia-2.8B to Llama-3-8B). BPO consistently improves win rates and entropy over DPO (Table 3, Figure 1) and achieves SOTA on AlpacaEval2 (Table 6).


## Weakness

One main motivation of BPO is to address the additional complexity introduced in f-PO. However, the theoretical optimality is based on the model having sufficient capacity, which is a vague expression. When the model is capable of completely capture the complexity of the preference data, different optimization criteria should not vary in terms of the final performance after convergence.

---

> ### Author Rebuttal · Authors · 2025-07-30
>
> We greatly appreciate reviewer **K9Vq**’s constructive feedback and recognition of the paper’s **theoretical guarantee**, **implementation simplicity**, and strong **empirical results**. We address the reviewer’s comments below.
>
> &nbsp;
>
> ---
>
> ### **Weakness**.
>
> *One main motivation of BPO is to address the additional complexity introduced in f-PO. However, the theoretical optimality is based on the model having sufficient capacity, which is a vague expression. When the model is capable of completely capturing the complexity of the preference data, different optimization criteria should not vary in terms of the final performance after convergence.  // What is the benefit of formulating policy optimization as density ratio estimation given sufficient model capacity?*
>
> &nbsp;
>
> ### **Response**.
>
> The optimality in Theorem 2 relies on the following key assumptions:
>
> - **A1.** The policy model $\pi_{\theta}$ has sufficient capacity to represent arbitrary distributions.
> - **A2.** The optimization is perfect. (i.e., it satisfies both **A2.1** and **A2.2**.)
>   - **A2.1.** The computation of the objective function is exact.
>   - **A2.2.** $\pi_{\theta}$ exactly minimizes the objective function. (This corresponds to the ‘argmin’ in Theorem 2.)
>
> These assumptions (**A1, A2.1, A2.2**) should be considered separately. Even if sufficient model capacity in **A1** is assumed to be satisfied, different outcomes can still occur depending on how well **A2** is satisfied.
>
> &nbsp;
>
> **[Comparison of complexity and theoretical benefits to f-PO]**
>
> The claim that BPO has lower complexity than f-PO specifically refers to the reduced burden of computing the exact objective function required to satisfy **A2.1**. Leveraging Theorems 2 and 3, we derive a simple closed-form expression (Eq. 10 of the manuscript) that enables $\pi_\theta$ to converge to the optimal policy.
>
> In contrast, f-PO requires a perfectly trained reward model $r_{\phi^*}$ and an infinite number of samples from the reference model to compute the partition function required to satisfy **A2.1**. This makes the exact objective computation significantly more complex and dependent on stronger assumptions. A key advantage of the density ratio estimation formulation in BPO is that it enables exact objective computation for **A2.1** without any additional assumptions or computational burden.
>
> Here is a summary of the requirements to satisfy **A2.1**.
>  - f-PO: Perfect reward model $r_{\phi^*}$, infinite number of samples from $\pi_{ref}$.
>  - BPO: Guaranteed by Theorems 2 and 3 with no additional assumptions required.
>
> &nbsp;
>
> **[Convergence under sufficient model capacity]**
>
> While sufficient model capacity ensures the ability to represent the optimal policy, achieving perfect optimization in **A2** still depends on the optimization dynamics. Different loss functions induce different gradients and optimization trajectories, and this affects the model's practical convergence point. In Eq. 11 of the manuscript, we mathematically analyze gradient behavior across various loss functions, and in Section 3.3, we present example analyses. These analyses highlight that loss design remains practically important, even with sufficient model capacity.
>
> &nbsp;
>
> ---
>
> We would like to once again thank the reviewer for their valuable feedback, and we would be happy to address any further questions or concerns.

---

> > ### Comment · Reviewer_K9Vq · 2025-08-05
> >
> > Thanks for the author's clarification of my question. I decide to keep my final rating.

---

### Official Review · Reviewer_5KtK · 2025-06-30

**Clarity:** 2
**Significance:** 3
**Originality:** 3
**Rating:** 5
**Confidence:** 4

**Summary:**

The authors propose a novel preference optimization algorithm designed around the bregman divergence. They formulate the preference optimization problem using the Bregman ratio matching framework (BPO), they show that this framework theoretically recovers the optimal policy when the preferences are modelled by the Bradley Terry model. The authors compare BPO to relevant prior literature and conduct experiments on preference optimization baselines and benchmarks exploring the practical properties of their objective.

**Questions:**

**Questions**

- The proof of proposition 1 is built around the Bradley Terry model, this model has some flaws for example it cannot model transitive preferences, how does this affect the objective you have designed?

- Please respond to the points raised in the weaknesses around clarity in the experiment section, what are the exact nature of the results in Table 3?

**Ethical Concerns:**

["NO or VERY MINOR ethics concerns only"]

**Final Justification:**

I stand by my initial score of 5, and recommend the paper is accepted. The authors have responded to the weaknesses and questions I raised, and conducted additional experiments in response to my recommendations.

**Limitations:**

yes

**Quality:**

3

**Strengths And Weaknesses:**

**Strengths**

- The paper proposes, to the best of my knowledge, a novel preference optimization algorithm. The algorithm is theoretically motivated as a generalization of DPO and the authors prove its optimality.
- The paper proposes a variety of different ways this generalization can be used, for example they derive a novel objective but also investigate how existing methods can be adapted using their method (Section 3.4 and Section 4.1.1).
- The work runs extensive experiments to empirically validate their claims.
- Their method outperforms a variety of relevant baselines approaches on standard baselines.

**Weaknesses**

I broadly like the paper and thus recommend accept, however I think the paper could be improved in the following ways:

Clarity in the writing:

- I found some aspects of the paper difficult to follow specifically the experiment section, in Table 3 it is very unclear what the coloured superscript results are, in line 219 the authors reference these results directly as a ‘win-rate drop’ but it is unclear as to the exact measurement carried out here.
- A variety of results reference broader literature e.g. a key part of the author’s argument for using the data ratio relies upon the concrete scores and it’s completeness property further detail on these concepts provided in the appendices would be useful for rigorously understanding the argument.

Diversity metrics:

- It might have been nice to use a variant of the pass@k metric to analyse how the diversity and win-rate vary across BPO and different baselines.

---

> ### Author Rebuttal · Authors · 2025-07-31
>
> We sincerely thank the reviewer **5KtK** for the helpful feedback and for highlighting the **novelty**, **theoretical foundation**, **versatility with DPO variants**, and thorough **empirical validation** of our approach. We also appreciate that the reviewer generally finds the paper favorable and recommends acceptance. We address the specific suggestions in detail below.
>
> &nbsp;
>
> ---
>
> ### **Weakness 1.**
>
> *I found some aspects of the paper difficult to follow specifically the experiment section, in Table 3 it is very unclear what the coloured superscript results are, in line 219 the authors reference these results directly as a ‘win-rate drop’ but it is unclear as to the exact measurement carried out here. // Please respond to the points raised in the weaknesses around clarity in the experiment section, what are the exact nature of the results in Table 3?*
>
> &nbsp;
>
> ### **Response to Weakness 1.**
>
> We agree that the explanation of the superscripts was insufficient. Since f-DPO, f-PO, and BPO are all extensions of DPO, we used the superscripts to indicate the relative performance gain (red) or loss (blue) compared to DPO as:
>
>
> - $ \text{Superscript}$ (%) = $\frac{\text{Performance} _{\text{Method}} - \text{Performance} _{\text{DPO}}}{\text{Performance} _{\text{DPO}}} \times 100$ %
>
> For example, the win rate of DPO (vs Preferred) is $48.5$, and the win rate of f-DPO (FKL) (vs Preferred) is $34.5$. The superscript is calculated as $\frac{34.5 - 48.5}{48.5}\times 100$ % = $-28.9$%, and we referred to it as a “win rate drop” in line 219. In the case of entropy, the value for DPO is $2.801$, and for f-DPO (FKL) it is $3.354$, which gives a superscript of $+19.7$%, computed as $\frac{3.354 - 2.801}{2.801}\times 100$ %. For BLEU, where lower values indicate better performance, the sign of the superscript is reversed accordingly.
>
> We basically adopted this notion from recent works such as **[56]**, where it is used to indicate how much a method improves over DPO. In the camera-ready version, we will explicitly describe this calculation method in the caption of Table 3.
>
> &nbsp;
>
> **[56]** Junkang Wu, Yuexiang Xie, Zhengyi Yang, Jiancan Wu, Jinyang Gao, Bolin Ding, Xiang Wang, and Xiangnan He. β-dpo: Direct preference optimization with dynamic β. Advances in Neural Information Processing Systems, 37:129944–129966, 2024.
>
> &nbsp;
>
> ---
>
> ### **Weakness 2.**
>
> *A variety of results reference broader literature e.g. a key part of the author’s argument for using the data ratio relies upon the concrete scores and it’s completeness property further detail on these concepts provided in the appendices would be useful for rigorously understanding the argument.*
>
> &nbsp;
>
> ### **Response to Weakness 2.**
>
> We agree that the connection between ratio matching and distribution matching in our paper relies on the concept of *concrete scores* and their *completeness property*. While this idea may be familiar to the likelihood ratio estimation community **[31, 33]**, we acknowledge that citing the original paper **[31]**, where this notion of *concrete scores* was first introduced, in lines 115–116 of the main manuscript may not be sufficient for readers in the preference optimization community.
>
> The concrete score is defined as the ratio of probability masses between two different states (e.g., $y_w$ and $y_l$) of a discrete random variable. It can also be represented as a vector of ratios between a specific state $y$ and all possible $N$ states, as shown below:
>
> - $C_{\theta}(y) = \left[\frac{\pi_\theta (y_i)}{\pi_\theta (y)}\right] _{i=1}^N  -1.$
>
> The completeness property indicates that once the function $C_\theta$ is defined, the value of $\pi_\theta(y)$ are uniquely determined as:
>
> - $\pi_{\theta}(y) = \frac{1}{\Sigma_{i} {C_{\theta}(y)+1}}.$
>
> Therefore, matching the pairwise ratios between different states to the target ratios guarantees proper distribution matching. (This is described as Theorem 1 in **[33]**.)
>
> This concept is used in the proof of Theorem 2 in our paper, specifically in lines 898–900 of the main manuscript. The corresponding part can be written without relying on prior literature as follows. Once learned policy $\pi _\hat{\theta}$ and target optimal policy $\pi _{\theta*}$ has following relationship for all $(x, y, y_i)$:
>
>  - $\frac{\pi_\hat{\theta} (y_i|x)}{ \pi_\hat{\theta} (y|x)} = \frac{\pi_ {\theta*} (y_i|x)}{ \pi _{\theta*} (y|x)},$
>
> Then, for any $(x, y)$, the probability of $\pi_\hat{\theta}$ and $\pi_ {\theta*}$ are identical as:
>
> - $ \pi_\hat{\theta} (y|x) = \frac{1}{\Sigma_{i} { \frac{\pi_\hat{\theta} (y_i|x)}{ \pi_\hat{\theta} (y|x)}}} = \frac{1}{\Sigma_{i} { \frac{\pi_{\theta*} (y_i|x)}{ \pi_{\theta*} (y|x)}}} = \pi_{\theta*} (y|x ).$
>
> This is why ratio matching leads to distribution matching in our setting. We will add background on the concrete score in Section 2.2 and add the relevant proof in lines 898–900 of Theorem 2, so that the paper is more self-contained and accessible to readers unfamiliar with this literature.
>
> &nbsp;
>
> **[31]** Aaron Lou, Chenlin Meng, and Stefano Ermon. Discrete diffusion modeling by estimating the ratios of the data distribution. In Forty-first International Conference on Machine Learning, 2024.
>
> **[33]** Chenlin Meng, Kristy Choi, Jiaming Song, and Stefano Ermon. Concrete score matching: Generalized score matching for discrete data. Advances in Neural Information Processing Systems, 35:34532–34545, 2022.
>
> &nbsp;
>
> ---
>
> ### **Weakness 3.**
>
> *It might have been nice to use a variant of the pass@k metric to analyse how the diversity and win-rate vary across BPO and different baselines.*
>
> &nbsp;
>
> ### **Response to Weakness 3.**
>
> The metric pass@k considers a generation successful if at least one of the $k$ generated outputs is correct. It is widely used for evaluating both the quality and diversity of a model's outputs. More precisely, it is defined as follows **[C1]**:
>
> - $\text{pass@k} = 1 - \frac{\binom{n - c}{k}}{\binom{n}{k}},$
>
> where $n$ is the number of generated samples per prompt, $k$ is the number of samples selected to compute pass@k, and $c$ is the number of "correct" outputs among the $n$ samples.
>
> For the experiment corresponding to Table 3, we computed the metric for BPO and baselines: the basic ones, such as SFT and DPO, f-DPO (FKL), which demonstrated high diversity, and f-DPO ($\chi^2$), which showed a high win rate. (Due to high API costs, we provide a limited number of baselines for this.) We randomly sampled 96 prompts from the test set and set $n=10$. Outputs that beat $y_w$ in GPT-based evaluation were considered “correct.”
>
>
> | Method            | Winrate | pass@2 | pass@3 | pass@4 | pass@5 | pass@6 |
> |------------------|---------|--------|--------|--------|--------|--------|
> | SFT              | 0.335   | 0.441  | 0.512  | 0.558  | 0.590  | 0.614 |
> | DPO              | 0.485   | 0.623  | 0.688  | 0.729  | 0.757  | 0.777 |
> | f-DPO (FKL)        | 0.345   | 0.498  | 0.579  | 0.631  | 0.668  | 0.696 |
> | f-DPO ($\chi^2$) | 0.535   | 0.617  | 0.680  | 0.721  | 0.750  | 0.772 |
> | BPO (SBA)             | **0.570** | **0.673** | **0.731** | **0.764** | **0.786** | **0.801** |
>
> &nbsp;
>
> BPO outperformed all baselines at every value of $k$, which we attribute to its strong performance in both sample fidelity and diversity.
>
> &nbsp;
>
> **[C1]** Chen, Mark, et al. (2021). Evaluating large language models trained on code. arXiv preprint arXiv:2107.03374.
>
> &nbsp;
>
> ---
>
> ### **Question 4.**
>
> *The proof of proposition 1 is built around the Bradley Terry model, this model has some flaws for example it cannot model transitive preferences, how does this affect the objective you have designed?*
>
> &nbsp;
>
> ### **Response to Question 4.**
>
> We formulated our research problem as finding a generalized objective function that preserves the optimality, simplicity, and generality of the original DPO objective. Because of this, our method naturally follows the same target optimal policy as DPO and inherits the Bradley-Terry (BT) modeling assumption.
>
> As the reviewer pointed out, BT modeling cannot represent cyclic preferences (e.g., A ≻ B ≻ C ≻ A), which is a known limitation. Addressing such non-transitive structures has been explored as a separate line of recent work (e.g., **[C2]**). We believe the BT modeling still performs effectively in practice. At the same time, if more expressive preference models beyond BT can also be formulated from a likelihood ratio estimation perspective, this opens up a promising future direction to extend BPO.
>
> &nbsp;
>
> **[C2]** Zhang, Yifan and Zhang, Ge and Wu, Yue and Xu, Kangping and Gu, Quanquan. Beyond bradley-terry models: A general preference model for language model alignment. In Forty-second International Conference on Machine Learning, 2025.
>
> &nbsp;
>
> ---
>
> We sincerely appreciate the reviewer’s insightful comments, and we remain happy to clarify any additional questions or concerns that may arise.

---

### Official Review · Reviewer_qCve · 2025-07-02

**Clarity:** 3
**Significance:** 3
**Originality:** 3
**Rating:** 5
**Confidence:** 2

**Summary:**

In this paper, the authors introduce Bregman Preference optimization, a generalized form of DPO that modifies the standard RLHF objective to use a Bregman divergence term (a well-studied concept in information geometry) in place of KL-divergence (note that this follows other recent approaches, such as Alfano, et al. 2024. I would ask the authors to consider citing this). Very interestingly, they show from first principles (Eq. 7) how the optimal policy in RLHF can be expressed as a matching problem between the two data ratios in Eq. 8 (Theorem 2.) (their final loss $\mathcal{L}_{\text{BPO}}$ in Eq. 10 is a tractable variant of Eq. 9 based directly on Theorem 2 with the equivalence property provided in Theorem 3). A particular advantage of this two ratio formulation is that it avoids needing an expensive partition function as in Eq. 3 (while I find this to be a very interesting reformulation, it’s not clear whether this is a strong argument given that such a partition cancels out in the standard DPO. I would like to see the authors address this point).

DPO and other probabilistic variants of (f-DPO) follow as special cases of $\mathcal{L}_{\text{BPO}}$ by modifying the convex function `h` in the Bergman equation (Eq. 6) (the full derivation for DPO is provided in the Appendix A.5. I would encourage the authors to consider putting this in the main text, since it illuminates several misunderstands I had earlier in the paper that are not fully addressed by the form given in Table 2). They further derive several novel losses by further modifying `h` as shown in Table 2. To arrive at their main loss, they discuss (Sec. 3.3) issues wrt the gradient scaling behavior of these new losses and devise a gradient technique (SBA) to make the gradient scaling follow more closely that of DPO under the logistic loss.

Much of the experiments are conventional and compare systematically against DPO and f-DPO (using HH), as well as a wider range of DPO variants (using ultrafeedback and closely following the setup of SimPO). For HH, the main results are shown in Table 3, with their SBA variant showing large win-rates over the preferred outputs and the SFT-trained model as well as improvements over the diversity of the generation (I have some questions about the setup here that I include below). The main results for DPO variants are when in Table 6, with their model obtaining the highest aggregate win-rate of all approaches, including several SOTA approaches (based on this, the authors claim that their approach achieves a new SOTA for Llama-3-8B models).

[Alfano et al. 2024 Meta-learning objectives for Preference Optimization, Arxiv 2411.06568]

**Questions:**

--  Not being familiar with the "predictive entropy" metric used for measuring diversity in the first experiments, it's not clear how it works. Can you provide a clear intuition for what this is measuring and add it to the text?

-- I'm wondering what the motivation was for using HH given that it is known to contain significant (>20%) noise. Relatedly, why is computing the win-rate against the preferred output sensible in this case (e.g., as compared with comparing directly against vanilla DPO)?

-- Given that the partition function in Eq. 3 and standard formulation of RHLF cancels out in DPO, why is it so attractive that it's avoided in your formulation? A discussion of this would be helpful.

-- The numbers from Table 6 are repeated from the SimPO paper/repo and not fully reproduced, is this correct? Do you imagine that differences in your training setting might yield significantly different results? If so, this would be helpful to identify as a clear limitation.

**Ethical Concerns:**

["NO or VERY MINOR ethics concerns only"]

**Final Justification:**

My main concerns were addressed during the rebuttal. After this and carefully considering the feedback of other reviewers, I decided to keep my score high.

**Limitations:**

yes

**Quality:**

3

**Strengths And Weaknesses:**

## strengths

-- A very novel and interesting reformulation and generalization of DPO-style objectives from first principles via ratio matching and   Bregman divergence. This approach opens the doors to a wide range of novel DPO variants that will likely inform many follow up studies in this area.

-- A comprehensive evaluation of their new losses against DPO and probabilistic variants such as f-DPO, as well as other DPO variants. In both cases, their new approach yields significant improvements, which shows the empirical utility of their formulation.

-- The paper is well written and easy to follow in spite of its rather technical content.

## weaknesses

(I note that all weaknesses I note below are, to my mind, of a minor nature.)

--  While their experiments are conventional, they are limited in scale (llama-8B). The claim of SOTA, while carefully qualified in the text, must be taken with this in mind.

-- The motivations for experimenting with HH are not entirely clear. See below for specific questions.

-- This work is not the first to use Bregman divergence, as noted above. I would request that the authors to address this point directly and detail any important differences with this related work.

---

> ### Author Rebuttal · Authors · 2025-07-31
>
> We sincerely appreciate the reviewer **qCve** for the detailed feedback and for acknowledging that our work is **very novel**, shows **empirical improvement**, and is **well-written**. We also thank the reviewer for noting that the concerns raised are of a minor nature. Below, we address each of the comments in turn.
>
> &nbsp;
>
> ---
>
> ### **Weakness 1.**
>
> *While their experiments are conventional, they are limited in scale (llama-8B). The claim of SOTA, while carefully qualified in the text, must be taken with this in mind.*
>
> &nbsp;
>
> ### **Response to Weakness 1.**
>
> We agree that our experiments are limited to the LLaMA-3-8B scale due to resource constraints. As stated in Lines 279–280 of the manuscript, we have carefully qualified our claim within this scope: *“To the best of our knowledge, BPO achieves state-of-the-art LC performance among preference-optimized models based on LLaMA-3-8B-Instruct.”*
>
> As the reviewer also noted, our intention is not to claim universal SOTA, but rather to demonstrate strong performance under comparable conditions. We agree that it would be more appropriate to tone down the expression from “SOTA” to “outperforms baselines,” and we will make this revision in the camera-ready version.
>
> &nbsp;
>
> ---
>
> ### **Weakness 2.**
>
> *The motivations for experimenting with HH are not entirely clear. See below for specific questions. // I'm wondering what the motivation was for using HH given that it is known to contain significant (>20%) noise. Relatedly, why is computing the win-rate against the preferred output sensible in this case (e.g., as compared with comparing directly against vanilla DPO)?*
>
> &nbsp;
>
> ### **Response to Weakness 2.**
>
> We conducted HH experiments to compare different instances of DPO, f-DPO, f-PO, and BPO, selecting HH because it was used in the original DPO, f-DPO, and f-PO papers. As discussed in lines 211–216 of the manuscript, we observed clear and interpretable trends in the distributional behavior of f-PO and f-DPO (e.g., mode-seeking vs. mode-covering). Therefore, we considered HH a suitable benchmark, at least for comparisons involving f-PO and f-DPO. Moreover, a dataset with some noise can be a useful indicator of robustness in practical settings. Results on other datasets, such as TL;DR and UltraFeedback, are also included in the main paper.
>
> Even if the preferred output used for evaluation is noisy, it is fixed across the evaluation of each preference optimization method, so relative comparisons on this fixed set are still meaningful. We highlight that Table 3 includes comparisons against SFT responses as well.  In addition, we report experiments that directly measure win rates between SBA and other baseline methods on HH using Pythia-2.8B, summarized in the table below.
>
> We observe a clear trend: the stronger the baseline in the original comparison setting, the harder it is for SBA to outperform it. Still, SBA wins more than 50% of the time against all methods, indicating that SBA consistently outperforms the baselines. In the direct comparison with DPO, as referenced in the review, our method achieved a 56.5% win rate.
>
> | Anthropic-HH | $f$-PO (JS) | $f$-PO (RKL) | $f$-DPO ($\chi^2$)|$f$-DPO (FKL)|DPO|
> |:-|:-:|:-:|:-:|:-:|:-:|
> |BPO (SBA)|50.5%|57.5%|52.0%|73.0%|56.5%|
>
> &nbsp;
>
> ---
>
> ### **Weakness 3.**
>
> *This work is not the first to use Bregman divergence, as noted above. I would request that the authors to address this point directly and detail any important differences with this related work.*
>
> [Alfano et al. 2024 Meta-learning objectives for Preference Optimization, Arxiv 2411.06568]
>
> &nbsp;
>
> ### **Response to Weakness 3.**
>
> Thank you for sharing the related preprint. The main contribution of our paper lies in reformulating the preference optimization loss from a likelihood ratio estimation perspective, aiming to achieve simplicity, optimality, and generality. In this formulation, Bregman divergence serves a technical purpose to support this generalization. As a result, our use of Bregman divergence differs structurally from Alfano et al. (2024).
>
> We frame the full distribution matching between $\pi_{\theta}$ and $\pi_{\theta^*}$ as a ratio matching problem, resulting in the objective:
> - $D_h (R _{data}||R _{\theta})$.
>
> In contrast, Alfano et al. (2024) use Bregman divergence to replace the KL divergence between the policy and the reference distribution in RLHF regularization:
> - $-E_{\pi_{\theta}(y|x)}[r_{\phi*}(x, y)] + \beta D_h(\pi_{\theta}(y|x)||\pi_{ref}(y|x))$.
>
> When only the regularization term is modified, as in their case, the resulting optimal policy may deviate from that of DPO (e.g., in f-DPO), and such formulations can offer less control over the overall training behavior compared to methods that extend the full objective.
>
> Moreover, while we prioritize simplicity, Alfano et al. (2024) focus on meta-learning by introducing a learnable function $h$, which increases training complexity. Therefore, our motivation, usage, and implementation of Bregman divergence are fundamentally different.
>
> We will briefly discuss this in the related work section of the main paper and provide technical details in Appendix B of the camera-ready version.
>
> &nbsp;
>
> ---
>
> ### **Question 4.**
>
> *Not being familiar with the "predictive entropy" metric used for measuring diversity in the first experiments, it's not clear how it works. Can you provide a clear intuition for what this is measuring and add it to the text?*
>
> &nbsp;
>
> ### **Response to Question 4.**
>
> We acknowledge that our explanation may have been insufficient, as the main manuscript cites previous works **[62, 54]** in line 209 that used the same predictive entropy metric, without providing further detail. Line 552 of Section C.4 in the supplementary material provides the exact reference for our actual implementation.
>
> Predictive entropy measures token-level diversity across generated sequences. Intuitively, it captures the average uncertainty of the model’s next-token prediction across different generations.
>
> Formally, the metric is computed by sampling $N$ sentences from the policy and measuring the entropy at each generation step $k$:
>
> - $\frac{1}{N} \sum_{i=1}^{N} \frac{1}{|y^{(i)}|} \sum_{k=1}^{|y^{(i)}|} H\left( \pi_\theta(\cdot \mid y^{(i)}_{<k}, x) \right)$
>
> - $= -\frac{1}{N} \sum _{i=1}^{N} \frac{1}{|y^{(i)}|} \sum _{k=1}^{|y^{(i)}|} \sum _{y_k^{(i)}=1}^{V} \pi _\theta\left( y_k^{(i)} \mid y^{(i)} _{<k}, x \right) \log \pi _\theta\left( y_k^{(i)} \mid y^{(i)} _{<k}, x \right)$.
>
> We will incorporate this explanation of the metric at line 206 in the main text for the camera-ready version.
>
> &nbsp;
>
> ---
>
> ### **Question 5.**
>
> *Given that the partition function in Eq. 3 and standard formulation of RHLF cancels out in DPO, why is it so attractive that it's avoided in your formulation? A discussion of this would be helpful.*
>
> &nbsp;
>
> ### **Response to Question 5.**
>
> As noted in lines 83–85 of the manuscript, policy models often converge to sub-optimal points due to limited capacity and imperfect optimization. Even with the same theoretical optimum, the final solution can depend on the shape of the loss function. Thus, extending the loss function can potentially yield diverse solutions and better performance, as is commonly pursued in loss generalization research.
>
> We analyzed different $h$ functions both theoretically and empirically via Proposition 4, Section 3.3, and Figure 5, showing that extending beyond DPO allows us to find instances with higher win rates and greater diversity.
>
> Table 1 of the manuscript summarizes the benefits of BPO as optimality preservation, simplicity, and generality for multiple objectives. Here, generality can be seen as a key difference between the DPO and BPO frameworks. Moreover, Table 1 clearly highlights the distinction between prior baselines ($f$-PO, $f$-DPO), which lacked optimality or simplicity in their generalization, and BPO, which satisfies all desiderata.
>
> &nbsp;
>
> ---
>
> ### **Question 6.**
>
> *The numbers from Table 6 are repeated from the SimPO paper/repo and not fully reproduced, is this correct? Do you imagine that differences in your training setting might yield significantly different results? If so, this would be helpful to identify as a clear limitation.*
>
> &nbsp;
>
> ### **Response to Question 6.**
>
> We faithfully followed the training and evaluation settings provided in the SimPO repository. In particular, we used the same versions (e.g., `alpaca-eval==0.6.2` and `vllm==0.5.4`) and decoding parameters, which are known to significantly impact SimPO's regeneration results, as we already described in our supplementary material (Lines 538–540).
>
> In our initial experiments, we closely reproduced the reported performance of SimPO as shown in table below. Based on this, we deemed it reasonable to reuse the baseline results from the original SimPO paper, and we clearly stated this in Lines 261–263 of the manuscript: *“Baseline results are taken from SimPO and f-PO.”* We acknowledge that fully reproducing all baselines would be ideal. In the camera-ready version, we will clarify the origin of all reported numbers more explicitly in the caption of Table 6.
> | Model                              | AE-LC ($\uparrow$) | AE-WR ($\uparrow$) |
> |--|--|--|
> | Mistral-7B-Base-SFT-SimPO         | 21.5          | 20.8          |
> | → Our reproduction                     | 20.9         | 20.4         |
> ||||
> | Llama-3-8B-Instruct-SimPO-v2      | 53.7          | 47.5          |
> | → Our reproduction                       | 53.1         | 46.8         |
>
> &nbsp;
>
> ---
>
> ### **Suggestion 7.**
>
> *Adding full derivation for BPO recover DPO (currently in Appendix A.5.) in the main text for understanding.*
>
> &nbsp;
>
> ### **Response to Suggestion 7.**
>
> Yes, in the camera-ready version, we will clearly explain in the main text how a specific choice of the function $h$ in BPO recovers the standard DPO formulation.
>
>
> &nbsp;
>
> ---
>
> We appreciate the reviewer’s feedback and are happy to clarify further if needed.

---

### Official Review · Reviewer_kev6 · 2025-07-03

**Clarity:** 2
**Significance:** 3
**Originality:** 3
**Rating:** 4
**Confidence:** 4

**Summary:**

This paper formulates the alignment problem as a ratio matching problem and proposes a generalized DPO loss, of which standard DPO is a special case. The authors introduce the use of Bregman divergence in this framework and propose a new scaled version of Basu's power divergence. Experiments are conducted to compare the proposed method with $f$-DPO and $f$-PO, demonstrating improved performance over baselines in terms of both win rate and response diversity.

**Questions:**

a. Table 3 reports win rates comparing alignment methods against SFT and preferred responses. Could you also provide the win rates between different alignment methods themselves? For example, how does JS Loss compare to SBA?

b. The results on alignment benchmarks in Table 6 are based on LLaMA-3-8B-Instruct, which has already undergone RLHF training. How does the proposed method perform when applied to base models without RLHF?

c. What is the performance of the proposed method on reasoning tasks such as MMLU, GSM8K, etc.?

I am open to raising my score if my concerns are addressed well.

**Ethical Concerns:**

["NO or VERY MINOR ethics concerns only"]

**Final Justification:**

The rebuttal addresses my concerns well and I am inclined to accept.

**Quality:**

3

**Strengths And Weaknesses:**

Strengths:

a. The paper offers a novel and insightful perspective by formulating DPO as a ratio matching problem and introducing a new family of generalized DPO losses using Bregman divergence.

b. Empirical results show that the proposed loss achieves both higher win rates and greater response diversity compared to baseline methods.

Weaknesses:

a. Table 3 only reports comparisons between generated responses and preferred/SFT responses. It would be valuable to include direct comparisons between BPO and $f$-PO variants, such as SBA vs. JS.

b. The results on AlpacaEval2 and Arena-Hard are only based on LLaMA-3-8B-Instruct, which has already undergone RLHF training. It remains unclear how well BPO performs when applied to base models without prior RLHF training.

c. As argued in previous RLHF literature, alignment methods may negatively impact a model’s reasoning capabilities or its ability to generate accurate responses, a phenomenon known as alignment tax. To assess this potential drawback, it is important to present results on reasoning benchmarks such as GSM8K and MMLU to determine whether the proposed method introduces such negative effects.

d. The writing could be further improved. For instance, in Section 3.1, terms like “concrete score” and “completeness property” are introduced without prior definition or context, which may confuse readers unfamiliar with the related literature.

---

> ### Author Rebuttal · Authors · 2025-07-31
>
> We sincerely appreciate the reviewer **kev6** for valuable feedback and recognition of **novelty** and **improved performance** regarding win rate and response diversity. We address the reviewer’s comments below.
>
> &nbsp;
>
> ---
>
> ### **Weakness 1.**
>
> *Table 3 only reports comparisons between generated responses and preferred/SFT responses. It would be valuable to include direct comparisons between BPO and $f$-PO variants, such as SBA vs. JS.*
>
> &nbsp;
>
> ### **Response to Weakness 1.**
>
> We agree with the reviewer that including direct comparisons between BPO and baseline methods (e.g., SBA vs. $f$-PO (JS)) can provide clearer insights into relative performance.
>
> To address this, we compared SBA against $f$-PO (JS) and $f$-DPO ($\chi^2$), which performed well in the comparisons against the preferred and SFT outputs. We also included comparisons against $f$-PO (RKL) and $f$-DPO (FKL), which showed weaker performance in those settings. Additionally, we included comparisons with vanilla DPO.
>
> The table below presents the win rate of BPO (SBA) over each baseline for Anthropic HH in the Pythia 2.8B setting. We observe a clear trend: the stronger the baseline in the original comparison setting, the more difficult it is to outperform. Still, SBA wins more than 50% of the time against all methods, demonstrating its consistent advantage.
>
> | Anthropic-HH | $f$-PO (JS) | $f$-PO (RKL) | $f$-DPO ($\chi^2$)|$f$-DPO (FKL)|DPO|
> |:-|:-:|:-:|:-:|:-:|:-:|
> |BPO (SBA)|50.5%|57.5%|52.0%|73.0%|56.5%|
>
> Additionally, the table below provides a direct win rate comparison between BPO (SBA) and other methods on the TL;DR dataset in the GPT-J setting. BPO also outperforms other methods in this setting, with a notably larger margin.
>
> | TL;DR | $f$-PO (JS) | $f$-DPO ($\chi^2$)|DPO|
> |:-|:-:|:-:|:-:|
> |BPO (SBA)|61.5%|64.5%|66.0%|
>
> &nbsp;
>
> ---
>
> ### **Weakness 2.**
>
> *The results on AlpacaEval2 and Arena-Hard are only based on LLaMA-3-8B-Instruct, which has already undergone RLHF training. It remains unclear how well BPO performs when applied to base models without prior RLHF training.*
>
> &nbsp;
>
> ### **Response to Weakness 2.**
>
> We agree with the reviewer that evaluating BPO on base models without RLHF is important for understanding its general applicability. Figure 7 in the main paper and Table 8 in the supplementary material already present experiments conducted on Mistral-7B-Base, which is a base model without RLHF. Additionally, we were able to run further experiments on LLaMA-3-8B-Base during the rebuttal period.
>
> The table below compares the performance of the base model with that of the model after being trained with SimPO and BPO. We closely followed the official SimPO repository to evaluate the base model, the model trained with SimPO, and the one trained with BPO. For the SimPO-trained checkpoint on Llama-3-8B-Base, the performance differed noticeably from the official reporting, so we report the performance of a re-trained checkpoint for this case. (For Mistral-7B-Base and LLaMA-3-8B-Instruct with SimPO, the reported results were nearly reproduced, and we therefore used the results from the original paper for comparison.)
>
> As a result of the comparison, we found that BPO outperforms SimPO on the given benchmarks, even when it is trained on base models without RLHF, such as Mistral-7B-Base (SFT) and LLaMA-3-8B-Base (SFT).
>
> | Method | AlpacaEval LC ($\uparrow$) | AlpacaEval WR ($\uparrow$) | Arena-Hard WR ($\uparrow$)|
> |:-|:-:|:-:|:-:|
> |Mistral-7B-Base (SFT) |8.4|6.2 |1.3 |
> |+ SimPO|21.5|20.8 |16.6 |
> |+ BPO|**23.7**|**20.9** |**16.9**|
> |||||
> |Llama-3-8B-Base (SFT)|6.2|4.6 |3.3 |
> |+ SimPO | 21.4|18.5|25.6|
> |+ BPO| **22.5**| **18.7**| **31.7**|
>
> &nbsp;
>
> ---
>
> ### **Weakness 3.**
>
> *As argued in previous RLHF literature, alignment methods may negatively impact a model’s reasoning capabilities or its ability to generate accurate responses, a phenomenon known as alignment tax. To assess this potential drawback, it is important to present results on reasoning benchmarks such as GSM8K and MMLU to determine whether the proposed method introduces such negative effects.*
>
> &nbsp;
>
> ### **Response to Weakness 3.**
> We evaluated the reasoning performance of LLaMA-3-8B-Instruct, as well as the SimPO and BPO models, both of which were fine-tuned from LLaMA-3-8B-Instruct. For this, we used the latest version of the popular `lm-evaluation-harness`  framework and tested it on several well-known reasoning benchmarks with the default settings. The table below shows the performance and ranking of each method on each benchmark.
>
> For the math benchmarks GSM8K and GSM-PLUS, both SimPO and BPO showed lower performance compared to the base model. While BPO performed better than SimPO on the math benchmarks, this drop, known as “alignment tax”, was especially noticeable. We believe this happened because the UltraFeedback dataset has very few math problems, which may have led to catastrophic forgetting of math skills.
>
> Since math problems have clear and verifiable answers, it may be better to combine with verifiable reward-based methods like GRPO **[A1]**. Another approach is using SFT loss together, as in ORPO **[A2]**, which is known to reduce forgetting and help keep the base model’s performance. These ideas might be useful when the goal goes beyond instruction-following to include more precise reasoning like math.
>
> Excluding math benchmarks, both SimPO and BPO showed performance improvements over the base model on GPQA, TruthfulQA, ARC, and MMLU. In three of these benchmarks, BPO achieved greater improvements, while SimPO performed slightly better than BPO on MMLU by a margin of 0.001. Overall, across these six benchmarks, BPO achieved the best average ranking of 1.5 among the base model, SimPO, and BPO.
>
> | Benchmark [shot]              | GSM8K [5] ($\uparrow$) | GSM-PLUS [5] ($\uparrow$)| GPQA [0] ($\uparrow$) | TruthfulQA [0] ($\uparrow$) | ARC [0] ($\uparrow$)| MMLU [0] ($\uparrow$)| avg_ranking ($\downarrow$) |
> |:------------------|:---------:|:------------:|:--------:|:----------:|:---------------:|:-------:|:--------:|
> | Llama-3-8B-Instruct | **0.760** (1) | **0.538** (1)   | 0.292 (3)   | 0.517 (3)      | 0.566 (3) | 0.637 (3) | 2.33         |
> | + SimPO            | 0.664 (3) | 0.457 (3)   | 0.310 (2)   | 0.639 (2)      | 0.576 (2) | **0.642** (1) | 2.17         |
> | + BPO              | 0.685 (2) | 0.461 (2)   | **0.328** (1)  | **0.652** (1)      | **0.601** (1) | 0.641 (2) | **1.50**     |
>
> &nbsp;
>
> **[A1]** Shao, Zhihong, et al. "Deepseekmath: Pushing the limits of mathematical reasoning in open language models." arXiv preprint arXiv:2402.03300 (2024).
>
> **[A2]** Hong, Jiwoo, Noah Lee, and James Thorne. "Orpo: Monolithic preference optimization without reference model." arXiv preprint arXiv:2403.07691 (2024).
>
> &nbsp;
>
> ---
>
> ### **Weakness 4.**
>
> *The writing could be further improved. For instance, in Section 3.1, terms like “concrete score” and “completeness property” are introduced without prior definition or context, which may confuse readers unfamiliar with the related literature.*
>
> &nbsp;
>
> ### **Response to Weakness 4.**
>
> We thank the reviewer for pointing out the lack of clarity regarding some terminology. We acknowledge that terms like *concrete score* and *completeness property* may be familiar in the likelihood ratio estimation literature, but could be unfamiliar to readers in preference optimization. We also agree that simply citing **[33]**, where these concepts were introduced (lines 115-116 of the manuscript), may not be sufficient to explain them clearly.
>
> The concrete score is defined as the ratio of probability masses between two different states (e.g., $y_w$ and $y_l$) of a discrete random variable. It can also be represented as a vector of ratios between a specific state $y$ and all possible $N$ states, as shown below:
>
> - $C_{\theta}(y) = \left[\frac{\pi_\theta (y_i)}{\pi_\theta (y)}\right] _{i=1}^N  -1$.
>
> The completeness property states that once the function $C_\theta$ is determined, the value of $\pi_\theta(y)$ are uniquely determined as:
> - $\pi_{\theta}(y) = \frac{1}{\Sigma_{i} {C_{\theta}(y)+1}}$.
>
> Therefore, matching the pairwise ratios between different states to the target ratios guarantees proper distribution matching. (This is stated as Theorem 1 in **[33]**.) We will provide a more detailed explanation of this preliminary in Section 2.2 of the camera-ready version.
>
> &nbsp;
>
> **[33]** Chenlin Meng, Kristy Choi, Jiaming Song, and Stefano Ermon. Concrete score matching: Generalized score matching for discrete data. Advances in Neural Information Processing Systems, 35:34532–34545, 2022.
>
> &nbsp;
>
> ---
>
> We sincerely thank the reviewer again for the thoughtful and constructive feedback, and we would be happy to further clarify any remaining questions.

---

> > ### Author Response · Authors · 2025-08-05
> > **Kind reminder regarding further discussion**
> >
> > Dear Reviewer **kev6**,
> >
> > We sincerely appreciate your thoughtful review and the constructive feedback you provided. We would like to kindly ask whether our responses have sufficiently addressed your concerns. If there are any remaining questions or issues, we would be pleased to provide further clarification.
> >
> > Best regards,
> >
> > The authors

---

> > ### Comment · Reviewer_kev6 · 2025-08-06
> >
> > Thanks for the detailed response and additional experimental results. I will increase my score.

---

### Note · Authors · 2025-08-15

We sincerely thank all reviewers, the AC and the SAC for their constructive feedback, and for recognizing the novelty, theoretical soundness, and strong empirical performance of our work on Bregman Preference Optimization (BPO).

### Key Contributions
BPO reformulates Direct Preference Optimization from a likelihood ratio estimation perspective, yielding a generalized framework that (i) preserves optimality under Bregman divergence minimization, (ii) maintains implementation simplicity, and (iii) encompasses a wide family of preference optimization objectives, including DPO as a special case. Our proposed SBA variant further addresses gradient scaling issues, leading to consistent gains in both win rate and diversity.

### For addressing reviewer concerns,

– Comparisons with other variants: We added direct head-to-head win rate tables comparing BPO (SBA) against strong baselines such as f-PO(JS), f-PO(RKL), and multiple DPO variants, confirming consistent >50% win rates.

– Performance on base models: We extended experiments to Mistral-7B-Base and LLaMA-3-8B-Base, showing BPO outperforms SimPO.

– Reasoning benchmarks: We evaluated GSM8K, GSM-PLUS, GPQA, TruthfulQA, ARC, and MMLU, finding BPO improves over both base and SimPO models.

– Clarity and definitions: We clarified definitions of concrete score and completeness property during the rebuttal and will add this background in the final version. We will also move key derivation for BPO recover DPO into the main text.

– Metric: We provided additional diversity evaluation via pass@k analysis. We will incorporate an explanation for predictive entropy.

– Relation to prior work: We explicitly contrast our use of Bregman divergence with Alfano et al. (2024), emphasizing our structural differences, focus on ratio matching, and reduced complexity.

Several reviewers noted that their concerns were fully addressed in our rebuttal, and some explicitly indicated an increased score.

We believe the paper now fully addresses the substantive points raised during review, with stronger empirical evidence, improved clarity, and theoretical positioning. We also emphasize that BPO’s ratio matching formulation and SBA scaling are model-agnostic and applicable to diverse LLM architectures and preference optimization settings.

We thank the reviewers again for their engagement and look forward to contributing this work to the community.

---

### Decision · Program_Chairs · 2025-09-17

**Decision:**

Accept (poster)

**Comment:**

The paper proposes Bregman Preference Optimization, a principled generalization of DPO from a likelihood ratio estimation perspective. The formulation preserves optimality under Bregman divergence, subsumes DPO as a special case, and introduces a family of objectives that can be implemented with comparable simplicity. The authors also propose a gradient scaling variant and demonstrate consistent improvements in both win rate and output diversity across multiple datasets and models, achieving strong results relative to recent DPO/f-PO variants.

Overall, reviewers found the work to be novel, theoretically sound, and empirically validated, with particular strength in the clarity of the framework and its strong empirical gains over baselines. Concerns raised included clarity in terminology, the limited scale of experiments, evidence of alignment tax on math reasoning tasks, and the need for more direct comparisons and positioning against related work. The rebuttal provided new experiments on base models and reasoning benchmarks, clarified terminology and metrics, and contrasted BPO with prior formulations, which reviewers agreed satisfactorily addressed their concerns.

I share the reviewers’ positive assessment: the paper makes a meaningful theoretical and practical contribution to preference optimization, and the rebuttal strengthened its empirical and conceptual positioning. I recommend acceptance.